# Structure of the human ATAD2 AAA+ histone chaperone reveals mechanism of regulation and inter-subunit communication

Carol Cho [1✉], Christian Ganser[2], Takayuki Uchihashi [2,3], Koichi Kato [2,4,5] & Ji-Joon Song [1✉]

ATAD2 is a non-canonical ATP-dependent histone chaperone and a major cancer target. Despite widespread efforts to design drugs targeting the ATAD2 bromodomain, little is known about the overall structural organization and regulation of ATAD2. Here, we present the 3.1 Å cryo-EM structure of human ATAD2 in the ATP state, showing a shallow hexameric spiral that binds a peptide substrate at the central pore. The spiral conformation is locked by an N-terminal linker domain (LD) that wedges between the seam subunits, thus limiting ATP-dependent symmetry breaking of the AAA+ ring. In contrast, structures of the ATAD2-histone H3/H4 complex show the LD undocked from the seam, suggesting that H3/H4 binding unlocks the AAA+ spiral by allosterically releasing the LD. These findings, together with the discovery of an inter-subunit signaling mechanism, reveal a unique regulatory mechanism for ATAD2 and lay the foundation for developing new ATAD2 inhibitors.

[1] Department of Biological Sciences, KAIST Stem Cell Center, Basic Science 4.0 Institute, and KI for BioCentury, Korea Advanced Institute of Science and Technology (KAIST), Daejeon 34141, Korea. [2] Exploratory Research Center on Life and Living Systems (ExCELLS), National Institutes of Natural Sciences, 5-1 Higashiyama, Myodaiji, Okazaki, Aichi 444-8787, Japan. [3] Department of Physics and Institute for Glyco-core Research (iGCORE), Nagoya University, Chikusa-ku, Furo-cho, Nagoya, Aichi 464-8602, Japan. [4] Graduate School of Pharmaceutical Sciences, Nagoya City University, 3-1 Tanabe-dori, Mizuho-ku, Nagoya, Aichi 467-8603, Japan. [5] Institute for Molecular Science (IMS), National Institutes of Natural Sciences, 5-1 Higashiyama, Myodaiji, Okazaki, Aichi 444-8787, Japan. ✉email: carol.cho@kaist.ac.kr; songj@kaist.ac.kr

The oncogene ATAD2 (ATPase family AAA domain-containing 2) is systematically upregulated in various cancers[1,2], and is associated with multiple oncogenic transcription factors such as cMyc and E2F[3]. Elevated levels of ATAD2 expression correlate with unfavorable patient prognosis in colorectoral[4], lung[1], and breast cancer[5], and knockdown of ATAD2 has been demonstrated to inhibit tumor proliferation and invasiveness[6]. Consequently, ATAD2 has been extensively explored as a therapeutic target for various cancers, resulting in the discovery of several potent small molecule ATAD2 inhibitors[7,8] with potentially more under development. Despite considerable interest in identifying ATAD2-targeted drugs, surprisingly little is known about the cellular functions of ATAD2 and the mechanism of tumor malignancy.

While ATAD2 was initially identified as a transcriptional co-activator of estrogen and androgen receptors[9,10], cMYC[3], and E2F[11] in human cancer cells, studies of yeast ATAD2 homologs suggest that ATAD2 is multifaceted, serving not only as a transcriptional regulator but also as a histone chaperone and chromatin boundary element[12]. Additionally, although many studies have associated yeast ATAD2 homologs with transcriptional regulation, it remains unclear whether yeast ATAD2 generally acts as a transcriptional activator or a repressor, or both.

Specifically, the budding yeast ATAD2 homolog Yta7 has been shown to regulate the expression of numerous genes including histone gene transcripts by either activating or repressing transcription[13–15], while the fission yeast ATAD2 homolog, Abo1, has been shown to function as a transcriptional repressor[16]. Some of the transcriptional roles of Abo1 and Yta7 were attributed to their histone chaperone activities, which directly affect nucleosome density and positioning. For example, deletion of Yta7 in budding yeast led to increased nucleosome density within genes[17], and catalytically active Yta7 in an in vitro reconstituted system decreased histone levels and disassembles chromatin, consistent with a role in gene activation[18]. In contrast, deletion of Abo1 in fission yeast caused a reduction in histone dosage and nucleosome occupancy[16], and recombinant Abo1 promoted histone assembly in a single-molecule nucleosome assembly assay[19]. Collectively, these studies indicate that Yta7 and Abo1 both function as histone chaperones, but that Yta7 promotes nucleosome disassembly while Abo1 promotes nucleosome assembly. In addition to their direct involvement in regulating nucleosome density, Yta7 and Abo1 also act as boundary elements that restrict the spread of silencing at promoters[13,14,20] and heterochromatin/euchromatin boundaries[21,22], as well as interacting with other histone chaperones such as Rtt106[13,14,20], FACT[13,16,22], and Scm3[23]. Furthermore, Yta7 promotes chromatin replication in an in vitro reconstituted replication system, suggesting that Yta7 is also involved in replication[18].

Building on such discoveries from yeast, studies in mammalian cells have begun to uncover the molecular relationship between human ATAD2 and chromatin. Consistent with roles as a histone chaperone, ATAD2 has been shown to function as a general facilitator of chromatin dynamics in human ES cells[24], and to regulate the HIRA and FACT histone chaperones by limiting their residence times on chromatin[25]. The potential importance of ATAD2 in chromatin replication has also been explored in a study showing the recruitment of ATAD2 to replication sites during S phase[26].

From a structural standpoint, ATAD2 exhibits a conserved multidomain architecture comprising an N-terminal acidic domain, two AAA+ ATPase domains, a bromodomain, C-terminal domain, and an exceptionally long C-terminal linker (CTL) that bridges the bromo- and C-terminal domain (Fig. 1a). In human ATAD2, only the structure of the bromodomain has been experimentally determined, exhibiting a canonical four-helical bundle structure consisting of αZ, αA, αB, and αC helices.

The ZA and BC loops form a hydrophobic pocket that coordinates acetyl-lysines of histone H4 tails[27,28], and explains the binding preference of ATAD2 for histones acetylated at H4 K5, K12, and K16[1,24,26]. The bromodomain structure has been vital in the development of currently available ATAD2 inhibitors, but constitutes only a small portion of the full protein.

Unlike most other histone chaperones that do not require ATP for activity, ATAD2 stands out due to the presence of an ATP-dependent AAA+ ATPase domain. The AAA+ domain adopts a conserved structural fold found in the AAA+ ATPase super-family, with each monomer consisting of a large α/β nucleotide binding domain (NBD) and a small α-helical bundle domain (HBD). These monomeric units oligomerize into hexameric rings where ATP binding pockets are formed at the interface of subunits and structural motifs from both subunits such as the Walker A/B, sensor I/II, and arginine finger motifs contribute to nucleotide binding and hydrolysis as reviewed in[29–31]. Although AAA+ ATPases share common structural and mechanistic features including dynamic symmetry breaking of rings, substrate binding and translocation through the central pore, and coordinated movement of AAA+ domains with substrate binding domains, the specific mechanisms employed by individual AAA+ ATPases diverge depending on variations in the structural core, substrate binding domains, and biological functions[32]. Consequently, different AAA+ ATPases have evolved different strategies to coordinate the activities of individual subunits within the AAA+ ring.

Our recent studies of Abo1 provided structural insights into the overall organization of an ATAD2 homolog, showing that Abo1 is a three-tiered hexameric ring where a bromodomain ring stacks on top of 2 AAA+ ATPase rings[19]. In these structures, we found that Abo1 AAA+ domains undergo dynamic nucleotide-dependent changes between symmetric rings and asymmetric spirals that alter central pore size and bromodomain arrangement, thus potentially regulating how ATAD2 binds histone substrates. We also discovered that Abo1 bound the histone H3 tail at the AAA+ ring central pore, which was crucial for Abo1-dependent nucleosome assembly. Despite such advances in understanding Abo1 structural mechanism, whether the divergent human ATAD2 homolog shares common structural features with Abo1 is an open question.

Here, we present cryo-EM structures of N-terminally truncated human ATAD2 (aa 403-1390) and its complex with the histone substrate H3/H4. We identify an N-terminal linker domain (LD) that wedges between AAA+ subunits at the hexameric spiral seam and potentially locks ATAD2 in a stable conformation. Comparison of the ATAD2 structures with and without bound H3/H4 suggests that histone H3/H4 binding causes the undocking of the N-terminal LD, potentially unleashing ATAD2 activity and triggering nucleotide-dependent dynamics. Furthermore, we also identify a gate loop that mediates inter-subunit communication in ATAD2, in parallel to inter-subunit signaling (ISS) motifs that have been discovered in other AAA+ ATPases. These results provide structural and mechanistic insights into human ATAD2 and suggest a unique auto-regulatory mechanism for a AAA+ ATPase. Moreover, our findings lay a structural foundation for developing cancer therapeutics targeting ATAD2.

## Results

**Cryo-EM structure determination of human ATAD2.** In our initial attempts to purify full length ATAD2, we encountered challenges including low protein yield, proteolysis, and severe aggregation under various conditions. However, by removing the acidic N-terminal domain to create a construct similar to the testis-specific short isoform of ATAD2 isoform (ATAD2-S)[1],

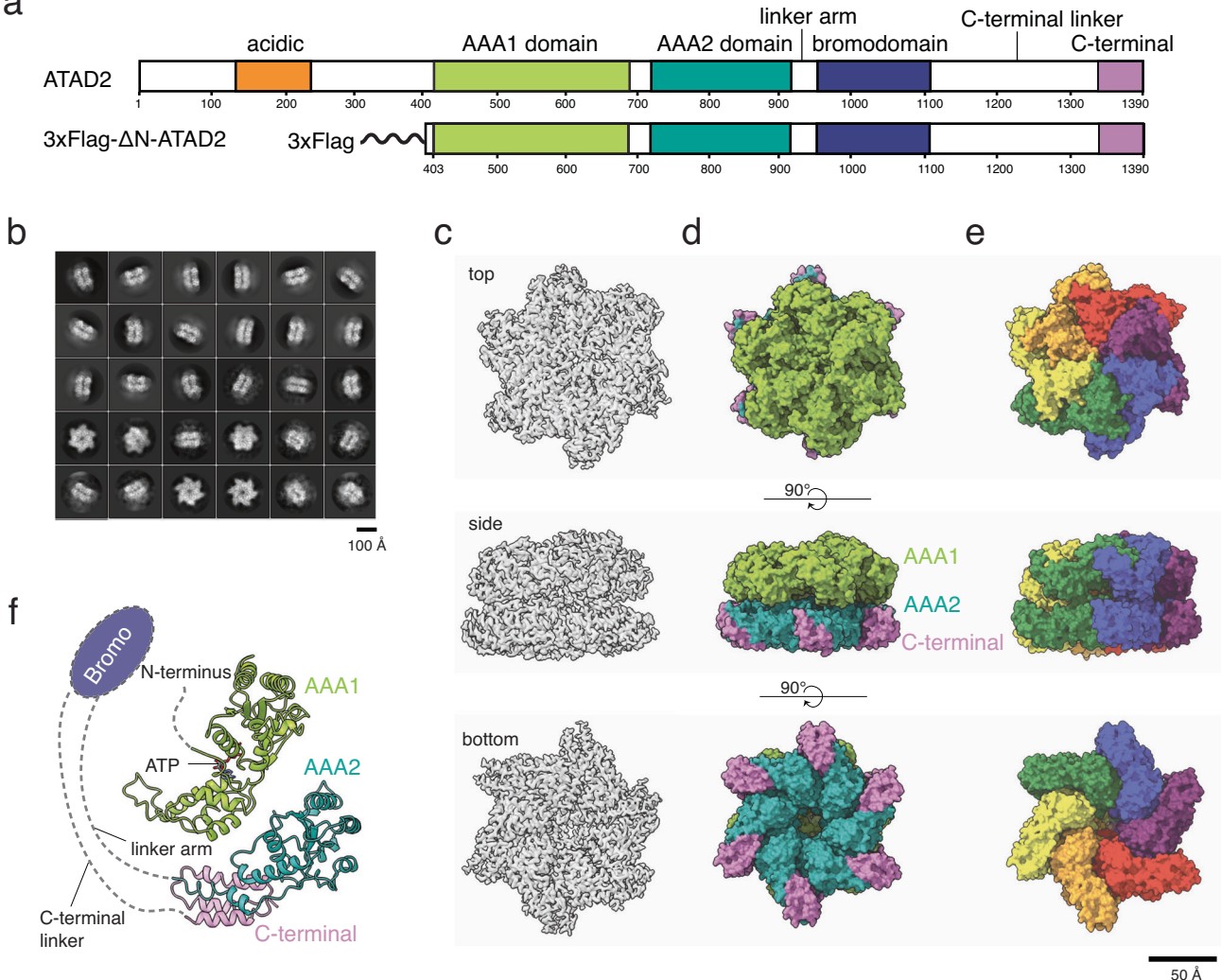

**Fig. 1 Cryo-EM structure determination of ATAD2. a** Primary structure of full-length ATAD2 and the tagged and truncated ATAD2 construct used for structure determination. **b** 2D class averages of ATAD2 Walker B mutant. **c**, **d** Top, side, and bottom views of the final sharpened density map of ATAD2 Walker B mutant hexameric ring contoured at σ = 5.5 (**c**), and color coded by domain with color scheme in Fig. 1a (**d**). **e** Surface representation of ATAD2 model colored by chain. **f** Schematic of ATAD2 monomer (chain A) with bound nucleotide, colored by domain. Dotted lines represent connectivity and approximate positions of domains that are unresolved in the cryo-EM map.

we were able to increase protein yields and homogeneity. Thus, in this study, we used N-terminally truncated ATAD2 (aa 403-1390, Fig. 1a) in parallel to our previous studies of Abo1, where we found that the acidic N-terminal domain was non-essential for in vitro function and overall protein stability[19]. Hereafter, we refer to N-terminally truncated ATAD2 as "ATAD2" or "wild type ATAD2", and any point mutations introduced are in the N-terminally truncated ATAD2 (aa 403-1390) background.

Unlike Abo1, which forms stable hexamers regardless of nucleotide state[19], wild type ATAD2 displayed a heterogeneous distribution of oligomeric assemblies both in the absence and presence of ATP, as determined by gel filtration, native gel electrophoresis, and negative stain electron microscopy (Supplementary Fig. 1). This suggested different characteristics of ATAD2 compared to Abo1. In addition, the catalytic ATPase activity of wild type ATAD2 was measured to be 0.03ATP/hexamer/min, a rate ~3000 times slower than Abo1 and close to the detection limit of our ATPase assays, implying that truncated ATAD2 did not maintain stable hexamers with functional nucleotide pockets. However, when a specific mutation (E532Q) which specifically blocks ATP-hydrolysis was introduced into the

Walker B motif, ATAD2 formed stable hexamers with homogeneous particle distributions (Supplementary Fig. 1).

Since Walker B mutants of many AAA+ ATPases have been shown to reflect the wild type conformation of AAA+ ATPases in an active ATP state[33–35], we determined the cryo-EM structure of the ATAD2 Walker B mutant. Specifically, ATAD2 Walker B mutant protein was purified by sequential affinity, ion exchange, and size exclusion chromatography, and the peak fractions of the size exclusion corresponding to hexameric size (~670 kDa) were vitrified on cryo-EM grids in the presence of ATP. Cryo-EM micrographs were collected using a Titan Krios 300 kV microscope with a Gatan Bioquantum GIF/K3 detector, and image processing was performed with Relion 3.1[36]. In summary, ~2 million particles were picked and subjected to multiple rounds of 2D class averaging (Fig. 1b) and 3D classification. The final cryo-EM map was generated with 212,925 particles to a resolution of 3.1 Å using the 0.143 FSC cutoff criteria (Fig. 1c, Table 1, and Supplementary Fig. 2).

The cryo-EM map exhibited well-resolved side chains (Supplementary Figs. 3a and 4) and revealed an overall structure of two stacked hexameric spirals. Atomic models of six AAA+

**Table 1 Cryo-EM data collection and refinement statistics.**

| | ATAD2 Walker B mutant with ATP | ATAD2 Walker B mutant – H3/H4K5Q complex with ATP | | |
| --- | --- | --- | --- | --- |
| | | Class I | Class II | Class III |
| **Sample Preparation** | | | | |
| Grid type | Quantifoil Cu R1.2/1.3 300 | | | |
| Cryo-specimen freezing | Vitrobot IV | | | |
| **Data Collection** | | | | |
| Electron Microscope | Titan Krios (300 kV) | | | |
| Detector | BioQuantum GIF/K3 (Electron counting mode) | | | |
| Total electron exposure (e/Å$^2$) | 60 over 50 fractions | 58.9 over 50 fractions | | |
| Defocus range (µm) | 0.8-2.2 | 0.8-2.4 | | |
| Pixel size (Å$^2$) | 0.85 | 1.1 | | |
| **Processing program** | Relion 3.1 | Relion 4.0 | | |
| Micrographs | 9,187 | 4,961 | | |
| Initial/Final particles | 2,167,392 /212,925 | 4,013,324 /76,408 | 4,013,324 /63,298 | 4,013,324 /49,284 |
| Symmetry imposed | C1 | | | |
| FSC threshold | 0.143 | | | |
| **Resolution (Å)** | 3.14 | 3.79 | 4.34 | 4.29 |
| **Refinement program** | PHENIX | | | |
| **Model composition** | | | | |
| Nonhydrogen atoms | 25,354 | 25,282 | 21,008 | 25,354 |
| Protein residues | 3,168 | 3,156 | 2,621 | 3,156 |
| B factors (Å$^2$) | -95.5 | -102.4 | -97.3 | -100.1 |
| **R.m.s. Deviation** | | | | |
| Bond Length (Å) | 0.007 | 0.006 | 0.003 | 0.002 |
| Bond Angle (°) | 0.808 | 0.811 | 0.739 | 0.663 |
| **Validation** | | | | |
| MolProbity Score | 1.90 | 2.01 | 2.10 | 1.92 |
| Clash Score | 6.89 | 9.53 | 13.36 | 9.12 |
| Poor rotamers (%) | 0.11 | 0.04 | 0.04 | 0.00 |
| **Ramachandran Plot** | | | | |
| Favored (%) | 90.86 | 91.35 | 92.63 | 93.34 |
| Allowed (%) | 8.98 | 8.49 | 7.25 | 6.56 |
| Disallowed (%) | 0.16 | 0.16 | 0.12 | 0.10 |
| **Mask CC** | 0.88 | 0.79 | 0.76 | 0.79 |

referred to as "subunit P1") and bottom (herein referred to as "subunit P6") subunits with an offset of ~12 Å. Consistent with other AAA+ ATPases, the hexameric assembly was stabilized by packing of the small α-helical bundle domain (HBD) of one subunit against the large α/β nucleotide binding domain (NBD) of the adjacent subunit, with the nucleotide pocket formed at the interface of adjacent protomers[30].

Despite the general resemblance of ATAD2 to Abo1 and other AAA+ ATPase Walker B mutants in an ATP state[19,34,38,39], ATAD2 had an unusually shallow rise and a narrow seam resulting in a relatively closed and planar spiral (Fig. 2a, b) that was in fact more similar to the symmetric planar ring conformation of Abo1-ADP (Fig. 2c) or Abo1-apo rather than Abo1-Walker B-ATP. In particular, the bottom AAA2/C-terminal ring did not show a prominent asymmetry or seam (Supplementary Fig. 5a), indicating that the top (AAA1) and bottom (AAA2/C-terminal domain) rings are conformationally uncoupled and that movements in the AAA1 domain ring and the AAA1-AAA2 linker are mainly responsible for the helical rise in ATAD2 (Supplementary Figs. 5b, c). Furthermore, in contrast to Abo1 and other AAA+ proteins where the seam subunits are highly disordered[19,34,38,39], the P1 and P6 subunits of ATAD2 were well-ordered and displayed indistinguishable density compared to other non-seam subunits in the local resolution map (Supplementary Fig. 3b).

Further comparison of the seams of ATAD2 and Abo1 in the ATP state revealed several differences. The cleft (25 Å vs. 48 Å) and offset (12 Å vs. 35 Å) between the P1 and P6 subunits were much smaller in ATAD2 than Abo1 such that the P6 subunit AAA1-small helical bundle domain (HBD) packed against the P1 subunit AAA1 large nucleotide binding domain (NBD) in ATAD2, while the P6 AAA1 HBD packed against the P1 AAA2 large NBD in Abo1 (Fig. 2d and e). Additionally, ATAD2 lacked the interlocking knob-hole structure observed in Abo1 (Fig. 2f, g, and h). The knob-hole structure in Abo1 involved the insertion of a "knob" formed by the AAA2 α0-insert of one subunit into a "hole" formed by the linker arm and small helical bundle domains of the adjacent subunit[19]. In ATAD2, the linker arms and α0-insert knobs were disordered in all subunits, despite being conserved based on secondary structure alignments. The absence of the knob-hole structure in ATAD2 might partially explain why it does not form stable hexamers in the absence of nucleotide (Supplementary Fig. 1). Overall, these structural differences highlight the unique characteristics of ATAD2 compared to Abo1.

domains were built into the density maps where the AAA1 domains occupied the top-tier spiral, and the AAA2 and C-terminal domains occupied the bottom-tier spiral (Fig. 1d, e). Bromodomains, linker arms, and C-terminal linker domains were not identified in our maps, likely due to the high disorder and/or flexibility of these domains relative to the AAA+ domains. However, based on similarities with Abo1[19] and Yta7[37] and the presence of additional density near the top surface of the AAA+ ring in 2D class averages, we propose a schematic of ATAD2 domain organization where the bromodomains and C-terminal linker domains are positioned near the top surface of the AAA+ ring (Fig. 1f).

**Hexameric spiral assembly of ATAD2.** In the hexameric structure of ATAD2, the six subunits were arranged in a right-handed spiral with a rise of ~2 Å and a rotation of ~60 ° around the helical axis. This resulted in a "seam" between the top (herein

**Nucleotide state and subunit conformation of ATAD2.** To gain further insights into the principles of ATAD2 oligomeric assembly, we examined the structure for differences in bound nucleotide and individual subunit conformation. At the resolution of the cryo-EM map, we were able to ascertain the identity of the nucleotides in all six nucleotide binding pockets of the AAA1 spiral (Fig. 3a and b). Five binding pockets of AAA1 were occupied by ATP, while the binding pocket of P6 at the spiral seam (P1/P6 interface) was occupied by ADP. No nucleotides were found in the AAA2/C-terminal domain ring, consistent with the absence of key nucleotide binding and hydrolysis sequences in the Walker A and B motifs of AAA2. Although most nucleotide pockets contained ATP, there were variations among different subunits in pocket size, arginine finger orientation, and the position of a loop protruding from the adjacent subunit (Fig. 3b).

Further analysis of the individual subunits of ATAD2 (P1-P6) also revealed notable differences. Unlike the AAA2/C-terminal domains which were relatively rigid with minimal divergence among subunits, there were significant differences among

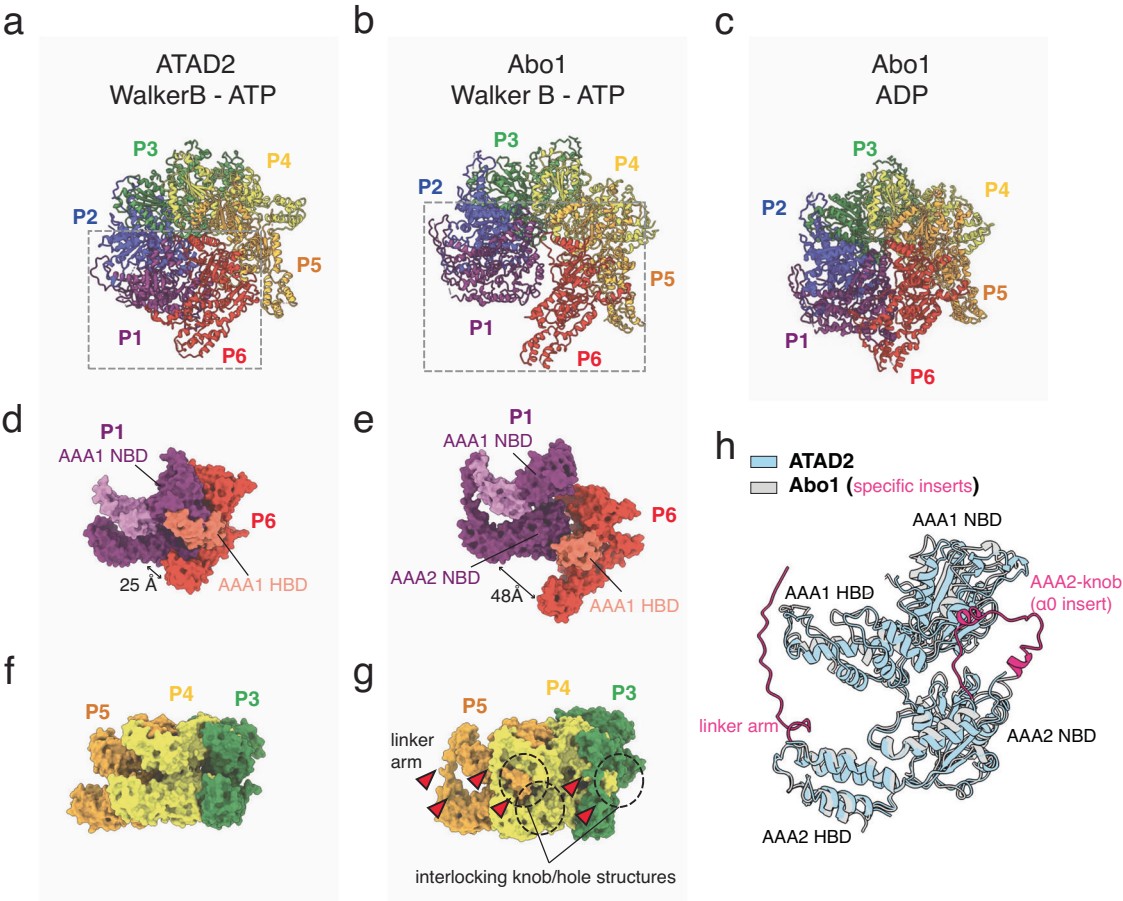

**Fig. 2 ATAD2 structure in comparison with Abo1. a–c** Hexameric structures of human ATAD2 Walker B mutant with ATP (**a**), Abo1 Walker B mutant with ATP (**b** PDB ID:6JQ0), and Abo1 with ADP (**c** PDB ID:6JPQ) colored by subunit (P1-P6). Dotted boxes represent spiral seam and are shown as isolated views in **d**, **e**. **d**, **e** Seam subunits P1 (purple) and P6 (red) from human ATAD2 Walker B mutant (**d**) and Abo1 Walker B mutant (**e**), showing differences in inter-subunit packing. (NBD: AAA+ nucleotide binding domain, HBD: AAA+ helical bundle domain). The AAA1 HBD of P1 and P6 are indicated with lighter hues. **f**, **g** Subunits P3-P5 of ATAD2 Walker B mutant (**f**) and Abo1 Walker B mutant (**g**). Interlocking knob-holes and linker arms are highlighted in (**g**) with dotted circles and arrowheads. **h** Superimposition of ATAD2 and Abo1 subunits. ATAD2 is shown in light blue and Abo1 in gray. The AAA2-α0 insert knob and linker arm of Abo1 are highlighted in magenta.

subunits in the AAA1 domains, where AAA1 moved as a rigid body with respect to AAA2 due to flexibility in the AAA1-AAA2 linker. As a result, the AAA1 subunits rose in height and rotated counter-clockwise (when viewing from the top of the AAA+ ring) with respect to the spiral axis (Supplementary Fig. 5 and Supplementary movie 1). In addition to the rigid body movement of the full AAA+ domain, there were local conformational differences in the helix α3-sheet β4 loop of AAA1 (Fig. 3c, d). This loop protruded out towards the neighboring subunit in subunits P3-P5 while it was retracted in subunits P1, P2, and P6 (Fig. 3b).

Interestingly, a similar structure termed the "nucleotide communication loop (NCL)" has been observed in the AAA+ ATPase Msp1[34], where melting of the NCL in response to ATP hydrolysis has been proposed to weaken inter-subunit contacts by disrupting interactions between the NCL and the N-terminal linker domain (LD). The same loop has also been proposed as a nucleotide gate loop in Yta7, closing the nucleotide pocket in the ATP state and opening in the ADP state. Whereas loop conformation directly correlated with nucleotide state in these structures, ATAD2 gate loop conformations did not directly correlate with nucleotide, but rather were dependent on subunit position within the hexamer, such that the three subunits close to the seam had open gate loops while the three subunits opposite

the seam had closed gate loops (Fig. 3d). This pattern of open and closed loops resembled the mixed conformation of inter-subunit signaling (ISS) motifs in a recent structure of p97 where loop conformation reflected the position of a subunit with respect to the post-hydrolysis (ADP or apo) subunit, and the engagement state of the pore loop with a substrate[40]. Specifically, in the closed gate loop conformation, L562, inserted into a hydrophobic groove of the adjacent subunit closing off the nucleotide pocket, while another residue, D560, stabilized the position of the two arginine fingers R586 and R589 (Fig. 3b). Although the primary sequence of the ATAD2 gate loop diverges from a conventional ISS motif that was originally defined by a conserved DGF tripeptide in membrane-bound AAA+ proteases[41] (Supplementary fig. 6), it exhibited a closed conformation similar to an ISS motif and likely functions in an identical manner.

**An N-terminal linker domain (LD) that locks the ATAD2 hexamer in a closed state**. When comparing individual ATAD2 subunits we also discovered a conserved N-terminal linker domain (aa406-423, referred to as "LD" hereafter, Supplementary Fig. 7). This structural element was only visible in the ADP-bound subunit, P6, and was sandwiched between the two subunits at the spiral seam (subunits P1 and P6, Fig. 4a and

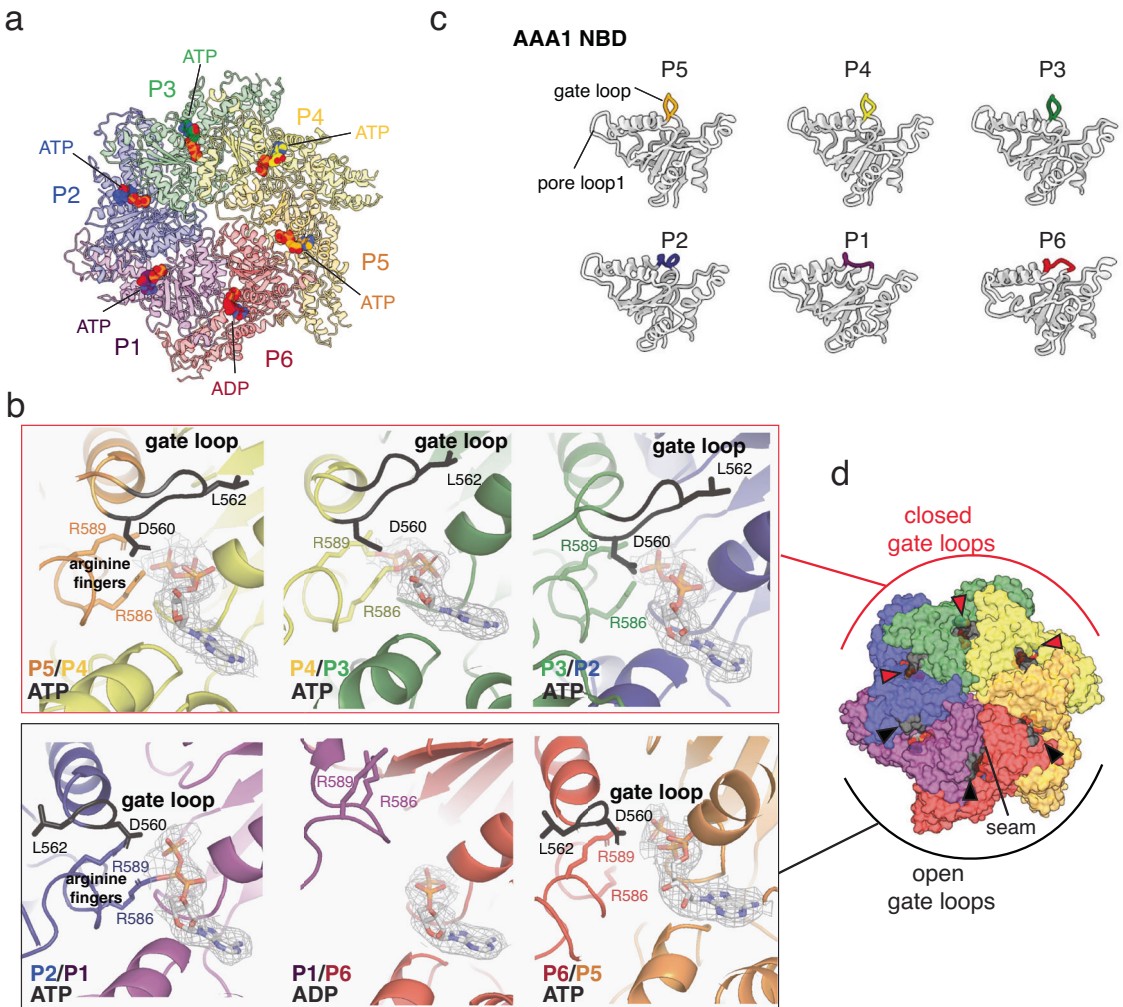

**Fig. 3 ATAD2 nucleotide pockets and subunits. a** Nucleotide pocket occupancy of hexameric ATAD2. **b** Closeup view of nucleotide binding pockets of ATAD2 showing nucleotide identity, gate loop position, and arginine fingers. Density map for nucleotide (contoured at σ = 4.5) is shown and arginine finger residues R586 and R589 are labeled. α3- β4 gate loops are labeled in black with side chains of conserved residues D560 and L562 shown. Density map for gate loops are shown in Supplementary Fig. 4d. **c** Comparison of α3-β4 gate loop conformation in ATAD2 subunits P1-P6. **d** Position of gate loops with respect to nucleotide binding pockets in the ATAD2 hexamer. Gate loops are colored black, with closed gate loops indicated with a red arrowhead, and open gate loops with a black arrowhead.

Supplementary Fig. 4b). Densities for an LD were not identified at any of the other subunit interfaces.

Structural alignments with other AAA+ ATPases revealed that the ATAD2 N-terminal LD occupied the same position as the N-terminal LD of the microtubule severing enzymes katanin[42] and spastin[43,44] (termed the "fishhook" element), and the membrane protein extracting enzyme Msp1[34] (termed the "LD linker domain") (Fig. 4b). In microtubule severing enzymes and Msp1, this domain consists of two helices (α0 & α1) connected by two linkers (L1 &L2) curved into a "fishhook-like" shape. It lies close to the initial substrate binding surface on top of the AAA+ spiral and has been proposed to mediate inter-subunit communication of nucleotide and substrate binding state.

In the ATAD2 structure, the N-terminal LD is composed of a loop corresponding to α1 and L2 of the katanin fishhook or Msp1 LD. Further comparison of N-terminal LD interactions with adjacent domains reveals similarities between katanin and ATAD2 (Fig. 4c). In both structures, the N-terminal linker interacts with the pore loop of the clockwise subunit, as well as helix α2 of the same subunit. However, the ATAD2 N-terminal LD has more extensive interactions that are absent in Msp1 and

microtubule severing enzymes, such as interactions with the arginine finger loop in the adjacent P1 subunit, and with helix α2 and sheet β1 in the same P6 subunit (Fig. 4d and e). The tight cleft formed by these interactions is also evident when comparing the distance from the N-terminal linker to the arginine finger loop of the adjacent subunit, which is significantly smaller in ATAD2 (6 Å) compared to in katanin (17 Å) and Msp1(14 Å).

In the ATAD2 structure, a salt bridge is formed between D415 of the N-terminal LD in P6 and R540 of pore loop 2 in P1, potentially stabilizing the P6 N-terminal LD-P1 interaction and hexamerization of ATAD2. To test this idea, we engineered D415A/R540A mutations in the Walker B mutant (E532Q) background. The resulting LD/Walker B triple mutant (D415A/ E532Q/R540A) displayed a broader size distribution with heterogeneous particles despite high expression levels and purity (Supplementary fig. 1). This suggests that the N-terminal LD might effectively lock the ATAD2 hexamer in a stable conformation, preventing ATP hydrolysis and symmetry breaking at another subunit. The well-ordered nature of the seam subunits in ATAD2 (Supplementary Fig. 3b), in contrast to other AAA+ proteins where the seam subunits are typically

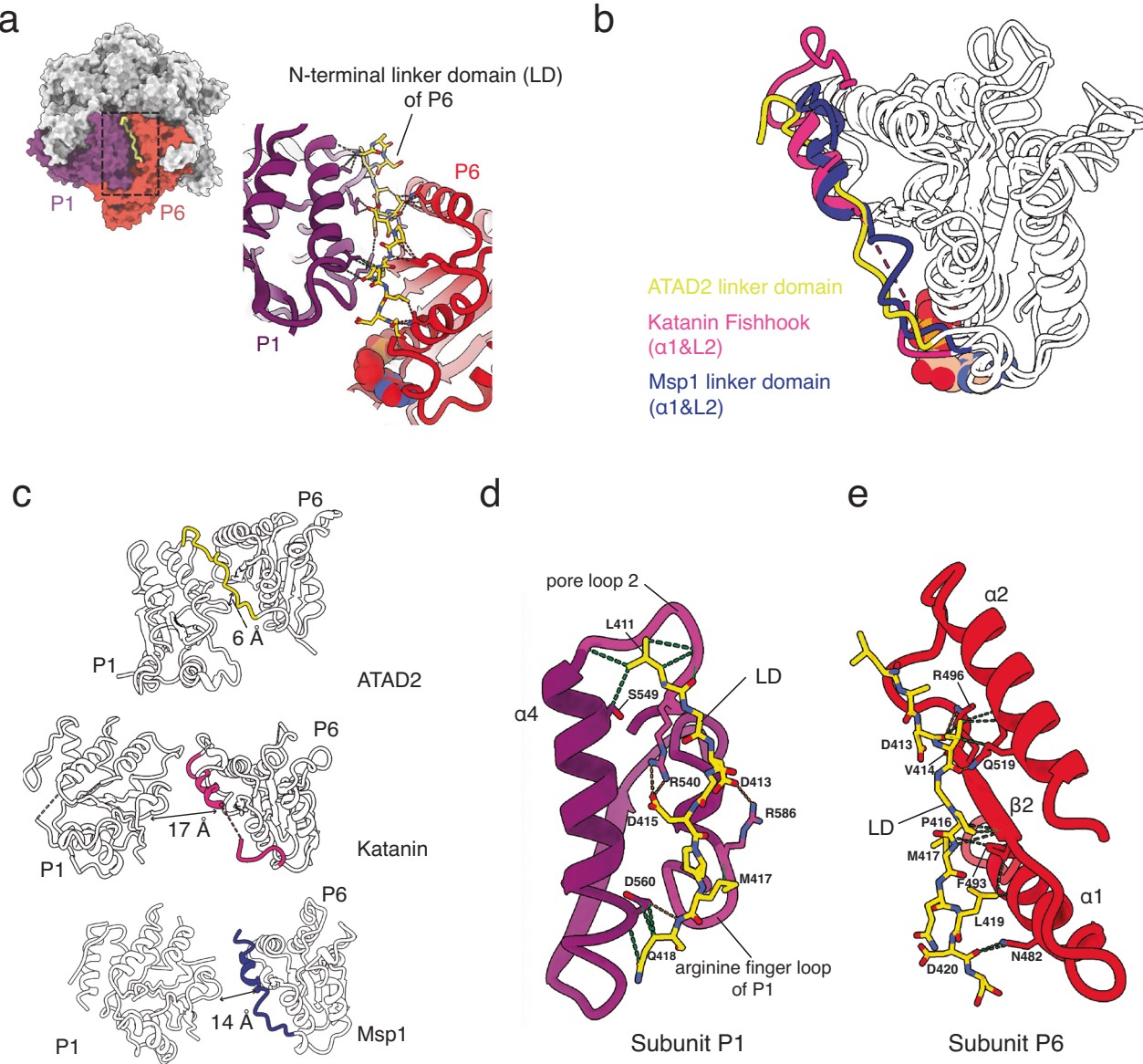

**Fig. 4 Structure of the ATAD2 N-terminal linker domain (LD). a** Position of the ATAD2 subunit P6 N-terminal LD (yellow) wedged between subunit P1 (purple) and P6 (red) in the hexameric spiral. (Density map of N-terminal LD is shown in Supplementary Fig. 4b) **b** Superimposition of the large nucleotide binding domains of ATAD2, katanin (PDB ID: 6UGD) and Msp1 (PDB ID: 6PE0) highlighting the position of the LD with respect to the nucleotide binding domain body in yellow, hot pink, and blue, respectively. **c** Comparison of distance from P6 LD to adjacent P1 subunit in ATAD2, katanin, and Msp1. Labeled distances are measured from middle of P6 LD α1 helix to P1 arginine finger. **d, e** Detailed hydrogen (orange) and van der Waals interactions (green) of the ATAD2 P6 LD with (**d**) the P1 subunit α4 helix, pore loop 2, and arginine finger loop (pore loop 2 and arginine finger loop highlighted in magenta), and (**e**) the P6 subunit α1, α2 helix, and β2 sheet.

disordered[38], further supports the role of the N-terminal LD in stabilizing the hexameric organizations. Although the functional effects of LD mutations on ATAD2 could not be determined due to the unavailability of a structurally homogeneous and catalytically active form of wild type ATAD2, we predict that disruption of the D415-R540 salt bridge might increase basal ATPase rates in wild type ATAD2.

**Pore loop and substrate binding of ATAD2.** After building an atomic model of ATAD2 into the density map, we also observed a region of additional density in the central pore of AAA1. This extra density was likely part of a substrate that co-purified with ATAD2 from Sf9 insect cells, as was also observed in our previous structure of Abo1[19]. The side chains of the substrate were not discernible in the density map, so we modeled the substrate as a linear peptide consisting of 5 alanines, with the N-terminus facing toward the interior of the pore, as seen in other AAA+ ATPases (Fig. 5a, left). The AAA1 pore loop1 of ATAD2 wrapped around the substrate with a spiral staircase-like configuration (Fig.5a, right), which is structurally well conserved in various AAA+ ATPases[38]. Consistent with this, ATAD2 AAA1 pore loop 1 W505 residues engaged the peptide substrate by inserting an orthogonally positioned aromatic ring between two side chains of the substrate.

However, compared to other AAA+ structures and even the homologous Abo1 structure, ATAD2 exhibited signs of a weakly bound peptide. First, the peptide substrate had weak density, allowing only modeling of 5 amino acids, in contrast to longer substrates of 14 or 22 amino acids in Abo1or translocating

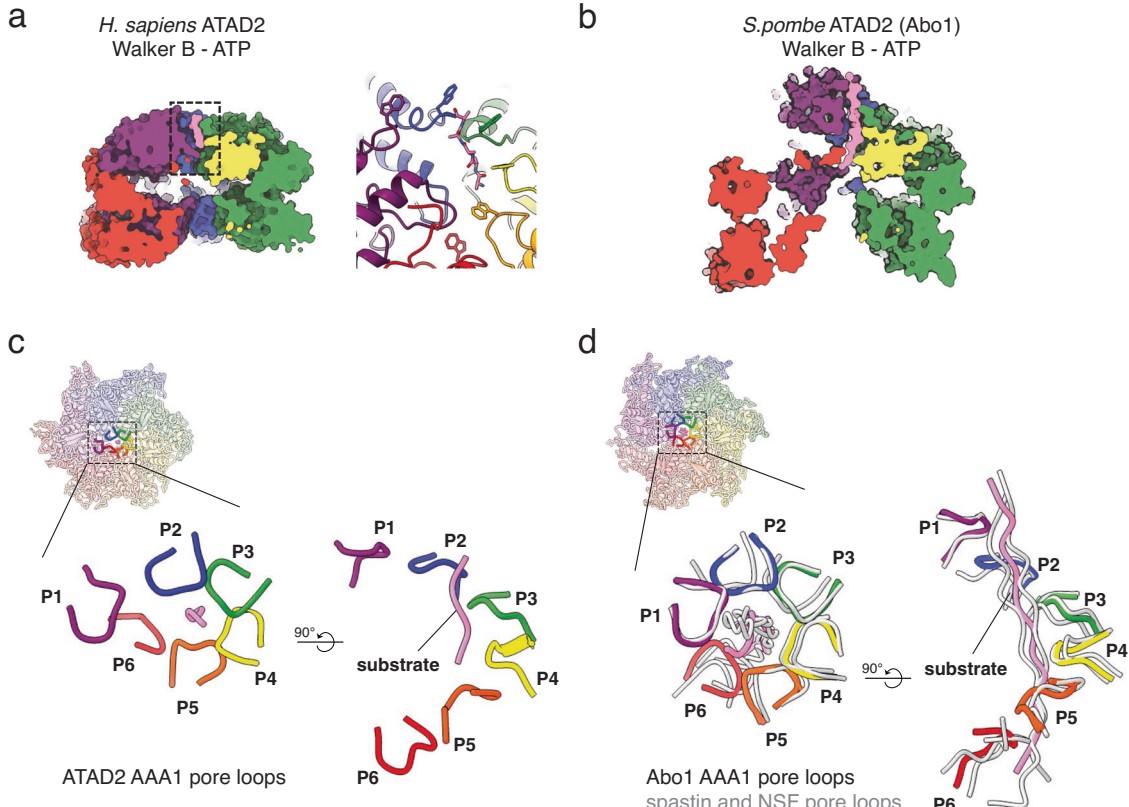

**Fig. 5 ATAD2 substrate binding at the AAA+ pore. a**, **b** Cut-open side view showing a substrate (pink) bound to the pore of ATAD2 (**a**) and Abo1 (**b**). Boxed region in (**a**) is shown in detail at right showing ATAD2 pore loop staircase with W505A surrounding a peptide substrate (density map for substrate is shown in Supplementary Fig. 4c). **c** Configuration of ATAD2 AAA1 pore loops 1 (colored by chain) surrounding a peptide substrate (pink) at the central pore of the ATAD2 hexameric ring. **d** Configuration of Abo1 AAA1 pore loops 1 (colored by chain, PDB ID: 6JQ0) surrounding a central peptide substrate (pink), superimposed with pore loops and peptide substrate of the AAA+ ATPases spastin (gray, PDB ID: 6PEN) and NSF (gray, PDB ID:6MDO).

unfoldases such as p97. Therefore, instead of a long stretch of peptide deeply inserted into the asymmetric spiral, only a short peptide was shallowly docked near the top surface of ATAD2 (Fig. 5a and b). Second, only AAA1 pore loop1 gripped the peptide, while AAA1 pore loop 2 and the pore loops of AAA2 did not participate in substrate interaction. Third, while translocating AAA+ ATPases usually have 5 to 6 subunits engaged with the substrate in the active state, only 3 ATAD2 subunits (P2-4) engaged with substrate, suggesting a "weak grip". The P1 and P6 subunits were significantly displaced from the conventional pore loop staircase (Fig. 5c, d), such as those seen in spastin[44] or NSF[45]. It is worth noting that the ATAD2 subunits involved in substrate interactions corresponded to the subunits with nucleotide pockets closed off by gate loops (Fig. 3e).

**Structure of the ATAD2-histone H3/H4 complex**. After understanding the structure of ATAD2, we next sought to investigate how ATAD2 interacts with its histone H3/H4 substrate. To obtain an ATAD2-H3/H4 complex, we incubated ATAD2 Walker B mutant and recombinant H3/H4 in a buffer with physiological salt concentrations, followed by chemical crosslinking with disuccinimidyl suberate (DSS). At the ATAD2:H3/H4 mixing ratios used, H3/H4 did not seem to affect the oligomeric state of ATAD2 as confirmed by native PAGE and negative stain EM (Supplementary Fig. 8). However, chemical crosslinking induced some higher-order oligomers which were also observed when crosslinking ATAD2 alone. To remove higher-order oligomers and over-crosslinking byproducts and identify only crosslinks in the

hexameric state of ATAD2, we purified ATAD2 Walker B-H3/H4 complex corresponding to hexameric ATAD2 by sucrose gradient fractionation.

XL-MS analysis revealed that ATAD2 crosslinked with histone H3/H4 at multiple positions through the AAA1, bromo- and C-terminal linker domains, while the AAA2 and C-terminal domains did not crosslink with H3/H4 (Supplementary Fig. 9a, b, and Supplementary data file 1). This implies that the top surface of ATAD2 with the bromo-and AAA1 domains forms the histone binding surface, while the AAA2 and C-terminal domains form an inert base on the bottom. All AAA1 residues crosslinking to histone H3/H4 mapped to the top surface of the AAA+ ring further supporting this idea (Supplementary Fig. 9c). Conversely, when examining the positions of histone H3/H4 crosslinks, it was observed that the N-terminal tails and histone bodies of H3 and H4 crosslinked to ATAD2. Notably, the N-terminus of H3 crosslinked to AAA1 pore loop, similar to Abo1, suggesting that the substrate observed in the ATAD2 structure is part of the histone H3 N-terminal tail.

Next, we performed cryo-EM on crosslinked and sucrose gradient purified ATAD2 Walker B mutant-H3/H4 complexes. We used wild type H3/H4 and a H3/H4K5Q mutant (a mimic of H4K5 acetylated H3/H4) as it has been shown that the ATAD2 bromodomain preferentially recognizes H4K5-acetylated histones[1,24]. The overall structures of both complexes were similar, but the resolution of the ATAD2-H3/H4K5Q complex was higher. Therefore, only the structure of the ATAD2-H3/H4K5Q complex is presented here. 4961 cryo-EM movies of the ATAD2 Walker B mutant-H3/H4K5Q complex in an ATP state

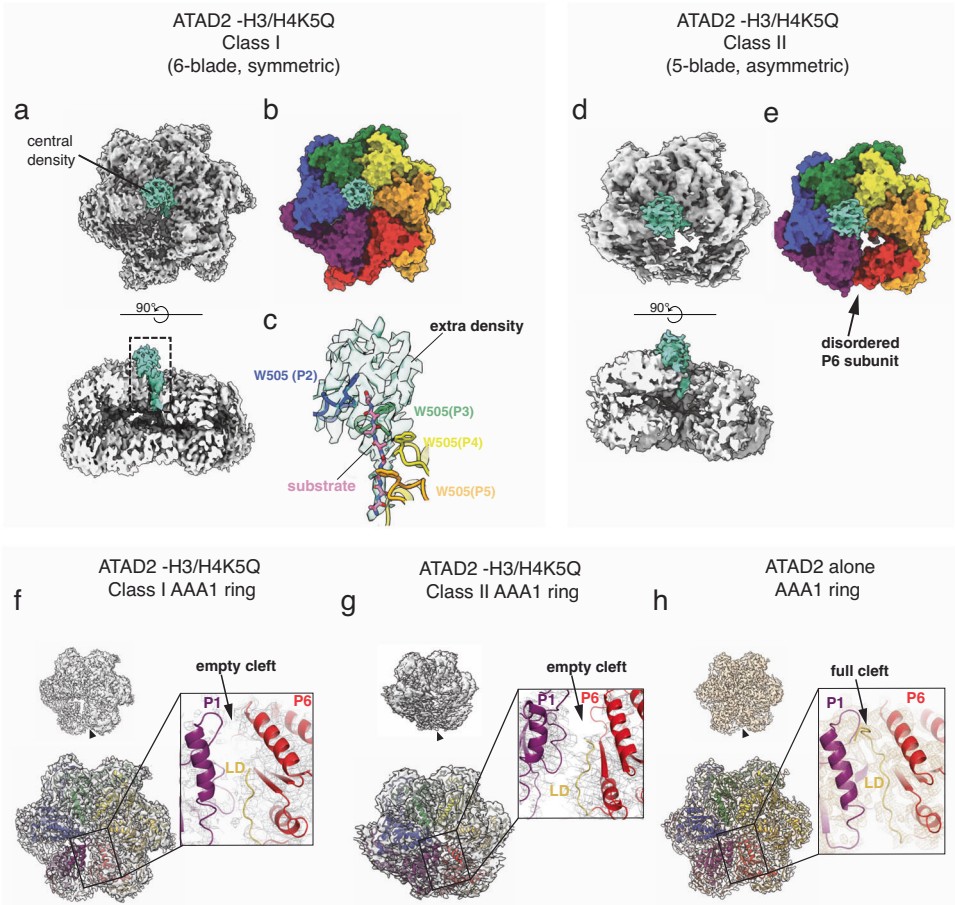

**Fig. 6 Asymmetric structures and substrate binding of the ATAD2-H3/H4 complex. a** Density map of ATAD2 Walker B mutant-H3/H4 complex class I (contoured at $\sigma = 3.5$). Extra central density after subtraction of AAA+ domains from map is colored in teal. **b** Surface representation of AAA+ domains (colored by subunit) overlayed with extra central density map (teal). **c** Detailed view of boxed region in (**a**), showing the central peptide substrate surrounded by the pore loop staircase, and extra globular density connecting to the peptide substrate. **d** Density map of the asymmetric structure of the ATAD2 Walker B mutant-H3/H4 complex class II (contoured at $\sigma = 3.5$). Extra central density after subtraction of AAA+ domains from map is colored in teal. **e** Surface representation of AAA+ domains (colored by subunit) overlayed with extra central density map (teal). P6 subunit is disordered in Class II structure, with no density for AAA+ helical bundle domains (HBD). **f–h** Density maps (contoured at $\sigma = 3.5$) and structures of AAA1 domains in ATAD2 Walker B mutant-H3/H4 complex class I (**f**), complex class II (**g**), and ATAD2 Walker B mutant alone (**h**). Arrowhead indicates seam (P1/P6 subunit interface). Boxed region shows detailed density map of seam and LD. LD model includes amino acids 408-423 in (**h**) and amino acids 414-423 in (**f**) and (**g**).

were collected on a Titan Krios 300 kV microscope with a Gatan Bioquantum GIF/K3 detector and processed with a similar pipeline to ATAD2 (Supplementary Figs. 10, 11 and Table 1). While processing of the ATAD2 structure yielded only a single major 3D class, the ATAD2-H3/H4 complex could be classified into three major 3D classes with resolutions of 3.8 Å (Class I), 4.3 Å (Class II), and 4.3 Å (Class III).

Class I exhibited the most symmetric structure, generally resembling the hexameric assembly observed in the ATAD2 structure (Fig. 6a and Supplementary Movie 2). Class III was similar to Class I but displayed an extra handle-like density that connected to one side of the AAA1 ring (Supplementary Fig. 12). This additional density was also observed as a weak signal in the 2D class averages (Supplementary Fig. 10). Although the identity of the extra density could not be definitively determined due to insufficient resolution, based on parallels to Abo1 and the general size and shape of the density, one possibility is that it corresponds to an ATAD2 bromodomain. Class II, in contrast to Class I and III, was distinctly asymmetric showing only 5 of the 6 blades of the hexameric ring due to disorder (Fig. 6d and Supplementary movie 3).

A notable feature common to all classes of the ATAD2-H3/H4K5Q complex was the presence of an extra globular density that was unobserved in the ATAD2 alone structure. This density was situated on the top surface of the AAA+ ring close to the central pore (Fig. 6a–e and Supplementary Fig. 12) and connected to a density (Fig. 6c) resembling the substrate peptide observed in the ATAD2 structure. The resolution of the globular density was insufficient to unambiguously determine its identity. However, together with the XL-MS data that indicates crosslinking of the H3 tail to the ATAD2 AAA1 pore loops and the connectivity of the globular extra density to the H3 tail, we favor the idea that the structure represents a single H3/H4 dimer docked close to the AAA+ ring by the H3 tail.

**Histone H3/H4-induced changes of the ATAD2 AAA+ ring**. In order to gain further insights into how histone H3/H4 binding might affect ATAD2 structure, we compared the AAA+ domain structure of the ATAD2-histone H3/H4K5Q complex to that of ATAD2. In agreement with XL-MS data that showed similar crosslinking patterns of ATAD2 and ATAD2-histone H3/H4, the

ATAD2 AAA+ domain models fit well into the maps of the ATAD2-H3/H4 complex without major rearrangements. This suggests that ATAD2 does not undergo large-scale conformational changes upon H3/H4 binding. In addition, the hexameric assembly, nucleotide occupancy, and gate loop conformations were overall similar in the presence and absence of H3/H4 (Supplementary Fig. 13), although in ATAD2-H3/H4K5Q Class II there was some ambiguity in the nucleotide identity at the P1/6 and P2/1 interface due to disorder.

However, prominent differences were observed at the seam (P1/P6), where the P1 exhibited slight disorder in Class I, and more pronounced disorder along with the P6 subunit in Class II (Fig. 6f–h and Supplementary Fig. 11). Consequently, there was no visible map density corresponding to amino acids 408 to 413 of the N-terminal LD of the P6 subunit and weak overall density for the LD from amino acids 413 to 423. This suggests that the N-terminal LD might be undocked from its original position sandwiched between P1 and P6, destabilizing interactions between seam subunits. This structural change might also be related to the overall lower resolution of the ATAD2-H3/H4 complex compared to ATAD2 alone, because the seam is no longer "fixed" by the N-terminal LD.

Superimposing the AAA+ domains from ATAD2 and ATAD2-histone complexes revealed that in the ATAD2-histone complexes, the P1 and P6 subunits moved away from each other, thus widening the seam cleft, and opening the AAA+ ring (Supplementary movies 2-4). The changes were most prominent in the asymmetric Class II conformation with less prominent changes in Class III and I. Consistent with potential movement of the N-terminal LD upon H3/H4 binding, examination of ATAD2 intramolecular crosslinks showed that the N-terminal LD crosslinked extensively with the nucleotide binding pocket, pore loop 1, bromodomain, and C-terminal linker (CTL) domain in the absence of histone H3/H4, but lost most of these crosslinks upon histone H3/H4 binding (Supplementary Figs. 9a, b). Together, these data support the idea that H3/H4 binding to the bromo- and CTL domain might allosterically release the N-terminal LD from the P1/P6 seam, open up the AAA+ ring, and prime the AAA+ ring for activation.

**ATAD2 symmetry breaking observed by HS-AFM.** To confirm whether histone H3/H4 might induce ATAD2 conformational changes as suggested by the ATAD2-H3/H4 cryo-EM structure, we employed high-speed atomic force microcopy (HS-AFM), a technique previously used to observe real-time ATP-dependent dynamics in Abo1[19]. In our previous work with Abo1, we observed robust ATP-dependent symmetry breaking that manifested as "disappearing blades" in HS-AFM movies due to large differences in subunit heights at the seam. While wild type Abo1 exhibited multiple cycles of symmetry breaking and recovery, Abo1 Walker B mutants displayed only a single cycle of symmetry breaking and were subsequently locked in the asymmetric conformation.

Similarly in ATAD2 Walker B mutants, we observed symmetry breaking events in which individual blades of the ATAD2 hexameric ring disappeared due to changes in subunit height (Supplementary Fig. 14a). This suggests that ATAD2 Walker B mutants undergo structural changes similar to Abo1, and likely reflect a conformational state of wild type ATAD2. However, unlike Abo1 Walker B mutants, where symmetry breaking events were irreversible in all observations, ATAD2 Walker B mutants occasionally reverted to symmetric states indicating subtle differences between ATAD2 and Abo1.

To assess the effect of histone H3/H4 binding on the frequency and reversibility of symmetry breaking in ATAD2 Walker B

mutants, we compared the behavior of ATAD2 in the absence and presence of histone H3/H4 (Supplementary Figs. 14b–d). Consistent with the open seam and asymmetric 5-blade conformation (Class II) of the ATAD2-H3/H4 K5Q complex observed by cryo-EM, we found that histone H3/H4 increased the frequency of symmetry breaking (quantified as symmetry breaking events per ring or per cumulative observation time) by ~40%, and the reversibility of symmetry breaking from 2% to ~30%. To investigate whether histone tails and histone H4K5 acetylation influence symmetry breaking, we also examined symmetry breaking events in the presence of tail-truncated ("tailless") histones and histone H3/H4K5Q. Both tailless H3/H4 and H3/H4K5Q stimulated ATAD2 symmetry breaking similar to wild type H3/H4. No statistically significant differences in symmetry breaking or reversibility were observed when comparing wild type H3/H4 with tailless H3/H4 or wild type H3/H4 with H3/H4K5Q indicating that the presence or modification of histone tails does not affect symmetry breaking. These findings collectively suggest that the H3/H4 body, rather than the H3/H4 tails, plays an important role in stimulating symmetry breaking of the hexameric ATAD2 ring.

## Discussion

**An autoinhibited state of ATAD2.** In this study, we present the cryo-EM structure of the human ATAD2 AAA+ ATPase, showing an architecture that follows the general assembly principles of AAA+ protein unfolding and disassembly machines. ATAD2 forms a shallow hexameric spiral staircase that binds a peptide, potentially the histone H3 tail, by conserved aromatic loops at the central pore.

However, several lines of evidence suggest that we have captured ATAD2 in a special autoinhibited state that has been previously unobserved in other AAA+ ATPases. First, in many AAA+ ATPases, breaking of ring symmetry to spirals has been shown to be essential for mechanical translocation of substrates and disassembly activity. Thus, subunits at spiral seams are disordered and variable in position as they have been caught in the act of translocation. In contrast, our ATAD2 structure has well-ordered subunits at the spiral seam that have indistinguishable resolution from other subunits (Supplementary fig. 3a) suggesting limited mobility and symmetry breaking. Second, we observe the N-terminal LD of the P6 subunit wedged between the seam subunits, where it effectively glues the P1 and P6 subunits in place, thus limiting subunit translocation and symmetry breaking. Third, only three ATAD2 pore loops maintain a weak grip on a short length of substrate in the central substrate, hinting at a non-productive or weakly-productive state if ATAD2 were to pull on substrates. Together, these lines of evidence imply that the N-terminal LD acts as a brake that maintains ATAD2 in an autoinhibited state thus preventing premature ATPase activation. Interestingly, the N-terminal LD was also proposed as a gate loop in recent structures of the ATAD2 yeast homolog, Yta7, based on its nucleotide-dependent conformational change near the nucleotide entry pocket[37]. Although the gate loop in Yta7 is not docked to seam subunits as in ATAD2 (discussed in more detail below), the idea of the N-terminal LD as a gating element that regulates nucleotide hydrolysis and exchange might be conserved among species.

**Symmetry breaking and activation of ATAD2 by H3/H4.** The structures of the ATAD2-histone H3/H4K5Q complex exhibited lower resolution than the structure of ATAD2 alone. This was likely due to the presence of multiple conformations that were more asymmetric and had increased disorder at the seam. These structural differences suggest that histone binding induces

symmetry breaking and AAA+ activation in ATAD2, leading to a conformation that is more similar to the open spiral structures of Abo1 and other AAA+ ATPases. This idea was also supported by HS-AFM observations showing increased symmetry breaking of ATAD2 in the presence of H3/H4 implying that H3/H4 might help prime ATAD2.

Concomitant with these conformational changes, the N-terminal LD was partially undocked and disordered in the ATAD2-histone complex in contrast to the well-ordered LD in the ATAD2 structure. This suggests that the binding of histone H3/H4 to ATAD2 might undock the LD through allosteric changes. XL-MS provided further clues to how histone H3/H4 binding might induce conformational changes in ATAD2. In the absence of histone, the N-terminal LD crosslinked not only with the AAA1 NBD as predicted by the cryo-EM structure, but also with the bromo- and C-terminal linker domains that are invisible in the cryo-EM structure. However, in the histone H3/H4-bound ATAD2 complex, many of these intramolecular crosslinks within ATAD2 disappeared, and were replaced with intermolecular crosslinks to histone H3/H4. Thus, we propose that histone binding to ATAD2 disrupts intramolecular interactions of the N-terminal LD with the AAA+ ring, thus triggering allosteric release of the N-terminal LD from the seam.

Overall, these findings indicate that H3/H4 binding to ATAD2 enhances symmetry breaking through an allosteric mechanism involving the LD. Consistent with the role of the LD proposed in other AAA+ ATPases such as Msp1[34], we propose that the N-terminal LD serves as a major regulatory domain where histone substrate binding status is communicated to the AAA+ domain to prime ATPase activity. Such a mechanism might be necessary to keep ATAD2 enzyme activity inhibited until recognition of an H3/H4 substrate.

**Histone H3/H4 binding mechanism of ATAD2**. Besides suggesting a mode of ATAD2 activation, the ATAD2-histone H3/H4 complex structure also provides insight into how ATAD2 binds its substrate. We confirmed that ATAD2 binds the histone H3 tail through the AAA1 pore loops, similar to Abo1 and Yta7. Moreover, an additional globular density connecting to the H3 tail was identified, suggesting that the H3/H4 body is positioned on the top surface of the AAA1 ring, with the H3 tail inserted into the central pore. In addition, extensive crosslinking between the H3/H4 body and the bromo- and CTL domains indicates that although these interactions are not directly visible in the cryo-EM structure, multiple weak interactions contribute to the binding of H3/H4 to ATAD2 in addition to binding of the H3 tail at the central pore.

Based on these findings, we propose a model in which one Abo1 hexamer binds one H3/H4 dimer, as a single pore can accommodate a single H3 tail. However, it cannot be completely ruled out that in addition to this binding mode, multiple H3/H4 dimers could also weakly bind to ATAD2 at multiple bromodomains.

**Comparison of ATAD2 homologs and implications for function**. Two major structural differences between human ATAD2 and its yeast homologs[19,37] emerge from this study, which suggests differences in their respective roles and mechanisms. Firstly, the absence of interlocking knob-hole interactions in ATAD2, when compared to Abo1 and Yta7 indicates that the hexamer formed by human ATAD2 is more dynamic and labile and can more readily disassemble into monomers. This is supported by our biochemical findings which show that wild type ATAD2 assumes a wide range of oligomeric forms, in contrast to Abo1 which remains a stable hexamer regardless of nucleotide

condition. Secondly, the N-terminal LD is observed to wedge between subunits in the ATAD2 structure but is unobserved in the Abo1 or Yta7 structures[19,37]. Although the LD may play a nucleotide gating role in all homologs, divergences in the N-terminal LD and the interacting bromo- and CTL domains, as well as differences in hexameric organization could explain why the LD acts as a brake only in ATAD2. It is also a possibility that these differences reflect differences in substrate binding preference (ATAD2 prefers acetylated histones[1,24,26] while Abo1 and Yta7 have no preference[46]) or biological function (promotion of nucleosome disassembly[18,24] vs. assembly[16,19]).

**An inter-subunit signaling (ISS) motif in ATAD2**. Another important finding of this study is the discovery of a gate loop in ATAD2 that corresponds to ISS motifs in other AAA+ ATPases. ISS motifs were discovered in classical clade m-AAA proteases as elements that transmit information about the nucleotide state to adjacent subunits and pore loops[41], and were originally defined by the presence of a crucial phenylalanine in a conserved DGF tripeptide constituting part of the α3 helix in the AAA+ NBD[41]. However, ISS motifs can diverge with a short insertion positioned C-terminal to the traditional ISS motif, such that in some cases like Msp1, it has even been termed a different name (the "nucleotide communication loop (NCL)"). Sequence-wise, the ATAD2 α3- β4 loop diverges from the originally discovered ISS motif, but shows similar conformations to the p97 ISS, where the loops jut out in a triangular shape towards the neighboring nucleotide binding pocket. Thus, it seems that the category of ISS motifs can be expanded to include a wider variety of sequences in the α3 C-terminus and α3- β4 loop than originally defined (Supplementary fig. 6).

**Limitations of the study and pharmacological implications**. Despite advances in understanding ATAD2 structure, a limitation of this study is the inability to obtain a catalytically active form of full-length wild type ATAD2. The truncated form of wild type ATAD2 used in this study lacks the ability to form and maintain stable hexamers (Supplementary fig. 1) with active nucleotide binding pockets, which might be due to the absence of specific post-translational modifications necessary for ATAD2 activation, or the exclusion of the ATAD2 N-terminus in our constructs. Supporting these ideas, ATAD2 has been shown to undergo extensive phosphorylation[18] and ubiquitylation at multiple sites[47–49], and in the case of Yta7, it has been shown that N-terminal phosphorylation is essential for catalytic activation[18]. Alternatively, an additional protein factor might be required to stabilize and activate the functional hexameric form of ATAD2. Thus, for now, a direct demonstration of ATAD2 function in vitro awaits testing.

Despite these limitations, structural insights gained from the ATAD2 Walker B mutant are likely applicable to the wild type ATAD2 mechanism, as has been shown to be true in many other AAA+ ATPases[33–35]. As the ATAD2 Walker B mutant represents a particular nucleotide state of wild type ATAD2, the general principles of histone binding, where the top surface of ATAD2 contributes to H3/H4 binding and the bottom surface serves as a rigid structural base, are also likely conserved in wild type ATAD2.

Our study lays the groundwork for designing structure-based inhibitors targeting the AAA+ domains of ATAD2 by elucidating two gating elements: the N-terminal LD and an ISS-like motif. Notably, recent structures of the AAA+ ATPase p97 demonstrated binding of the p97 inhibitor NMS-973 to the ISS motif[40], indicating that similar strategies could be applied to target ATAD2. Considering the variable sequences of ISS loops among

different AAA+ ATPases, our structural findings offer promising new strategies to target ATAD2.

## Methods

**Cloning and protein overexpression.** DNA encoding aa 403-1390 of ATAD2 was PCR amplified from the full-length codon-optimized synthetic ATAD2 gene (Genscript), and subcloned into an in-house modified pFastBac1 vector with an N-terminal 3xFlag tag and Tev protease cleavage site using EcoRI and XhoI restriction sites. Point mutants of ATAD2 (E532Q, D415A, and R540A) were created by inverse PCR with KOD plus polymerase (Toyobo) according to product manual instructions. Cloned plasmids were transformed into DH10Bac competent cells to produce bacmids that were subsequently transfected into SF9 cells to produce baculovirus, and baculovirus was amplified to P3 according the Bac-to-Bac Baculovirus Expression Kit (Thermo Fisher Scientific). ATAD2 proteins were expressed by infecting 950 mL of SF9 cells at a density of 2.5–3.5 ×10⁶ with 50 mL of P3 virus for 45-48 h. Recombinant *X. laevis* histones H3 and H4 were overexpressed in *E. coli* and purified by inclusion body purification as described by Luger et al[50].

**Protein purification of ATAD2 and histone H3/H4.** Harvested SF9 cells were resuspended in 20 mL of lysis buffer (50 mM Tris-HCl (pH 8.0), 300 mM NaCl, 5% glycerol) per liter cells, and lysed by 4 freeze/thaw cycles in the presence of 1 mM PMSF and cOmplete protease inhibitor cocktail tablet (Roche). Lysates were clarified by centrifugation at 39,000 xg for 1.5 h, and were batch bound to 1 mL of anti-DYKDDDDK G1 affinity resin (Genscript) per L cells for 2 h. Resin was washed with a sequence of 5 column volumes (CV) lysis buffer, 5 CV wash buffer 1 (50 mM Tris-HCl (pH 8.0), 500 mM NaCl, 5% glycerol), 5 CV wash buffer 2 (50 mM Tris-HCl (pH 8.0), 1000 mM NaCl, 5% glycerol), followed by 5 CV wash buffer 1, 5 CV wash buffer 2, and 5 CV lysis buffer. Proteins were eluted by incubating Flag resin with Flag elution buffer 1 CV at a time for 10 min (lysis buffer with 0.08 mg/mL DYKDDDDK peptide (Shanghai Apeptide)). A total of 5-7 CV was eluted and checked for yield and purity by SDS-PAGE. Peak fractions were pooled and diluted with an equal volume of no salt buffer (50 mM Tris-HCl (pH 8.0), 5% glycerol), and loaded onto a HiTrapQ 5 mL column (Cytiva Life Sciences). Fractions were eluted with a 50 mM to 1000 mM NaCl gradient over 20 CV. Peak fractions with a A260/280 ratio of <0.6 were pooled, concentrated with an Amicon Ultra-15 100 kDa cutoff centrifugal concentrator (Millipore), and run over a Superose 6 increase 10/300GL (Cytiva Life Sciences) column in 25 mM HEPES (pH 7.5), 250 mM NaCl, 5% glycerol, and 1 mM DTT. Gel filtration peak fractions corresponding to hexameric ATAD2 were pooled, concentrated and flash frozen until further use. Typical protein yields were on the order of ~100 ug purified protein per L cells.

Histone H3/H4 complex was prepared as described by Luger et al[50]. Briefly, *X. laevis* histones H3 and H4 were extracted in their denatured forms from bacterial inclusion bodies, and purified further by gel filtration and cation exchange chromatography. Purified H3 and H4 were dissolved in unfolding buffer (20 mM Tris-HCl (pH 7.5), 6 M Guanidine-HCl, and 5 mM DTT) and mixed at equimolar ratios, refolded by dialysis overnight in high salt buffer (10 mM Tris-HCl (pH 7.5), 2 M NaCl, 1 mM EDTA, and 5 mM β-mercaptoethanol), and purified over a Superdex 200 26/60 (Cytiva Life Sciences) column. Peak fractions corresponding to histone H3/H4 tetramer were pooled, concentrated, and used for ATAD2-histone H3/H4 complex formation.

**Preparation of ATAD2-histone H3/H4 complex.** To obtain ATAD2-histone H3/H4 complexes for XL-MS or cryo-EM data collection, a concentrated stock of refolded H3/H4 or H3H4K5Q stored in 2 M NaCl was first diluted into 150 mM NaCl buffer. Subsequently, ATAD2 Walker B mutant protein and refolded histone H3/H4 were mixed at a 1:1.5 molar ratio in a buffer consisting of 25 mM HEPES (pH 7.5), 150 mM NaCl, 5% glycerol, and 1 mM DTT, and incubated for 30 min on ice. To initiate cross-linking, 1 mM of DSS (Sigma-Aldrich) was added to proteins and incubated for 30 min at 37 °C, and quenched by the addition of ammonium bicarbonate to a concentration of 50 mM and an additional 20 min incubation at 37 °C. Crosslinked ATAD2-histone H3/H4 complex was further purified by a 10-30% sucrose gradient run for 17 h at 65,000 xg to remove over-crosslinked aggregates and excess histone H3/H4. Sucrose gradient fractions for cryo-EM data collection were analyzed by running on a NuPAGE Bis-Tris native 4-16% gradient gel (Thermo Fisher Scientific) at 150 V for 2.5 h.

**Cryo-EM grid preparation and data collection.** Before vitrification, proteins were buffer exchanged to 25 mM HEPES (pH 7.5), 250 mM NaCl, 0.025% beta-octyl-glucoside, 1 mM DTT, and 1 mM Mg-ATP and concentrated to 1.2 (ATAD2 Walker B-H3/H4K5Q complex) or 1.8 mg/mL (ATAD2 Walker B mutant). Proteins were frozen on glow-discharged Quantifoil Cu R1.2/1.3 300 mesh grids with a Vitrobot IV (Thermo Fisher Scientific) set at 15 °C and 100% humidity and blotted with -10 or 0 blotting force with 3 s blotting time. For the ATAD2 Walker B -ATP dataset, 9187 movies were collected using a Titan Krios 300 kV microscope with a Gatan Bioquantum GIF/K3 direct detector varying the defocus from 0.8 to 2.2 μm with 0.5 μm step size at Harvard Center for Cryo-Electron Microscopy (HC2EM) (Table 1 and Supplementary fig. 2). For the ATAD2 Walker B-histone H3/H4-ATP dataset, 4,961 movies were collected using a Titan Krios 300 kV microscope with a Gatan Bioquantum GIF/K3 direct detector varying the defocus of 0.8 to 2.4 μm with 0.2 μm step at POSTECH IMP.

**Cryo-EM data processing.** For the ATAD2 alone dataset, all processing was performed in Relion 3.1[36]. A total of 2,167,392 particles were from motion corrected (using MotionCorr2[51]) and CTF corrected micrographs (using CTFFIND4.1[52]) and subjected to multiple rounds of 2D class averaging (Supplementary fig. 2). Multiple rounds of 3D classification were performed starting with the 417,689 particles selected from 2D class averaging. The particles from the highest-resolution 3D classes were refined by 2 rounds of iterative refinement (3D auto-refine and CTF refine in Relion 3.1) without a specified reference mask. The refined model was sharpened by post-processing in Relion 3.1 using a low-pass filtered and extended mask, with a B-factor of -95.5. The final 3.1 Å resolution map was obtained based on 0.143 FSC cutoff criteria. Using the atomic model of Abo1 (PDB ID: 6JQ0) as an initial guide for backbone placement, residues of ATAD2 were built into the cryo-EM map using Coot[53] followed by real space refinement in Phenix[54].

For the ATAD2 Walker B -histone H3/H4K5Q complex dataset all processing was performed in Relion 4.0 (Supplementary fig. 10)[36]. A total of 4,013,324 particles were picked the template-free autopicking with a Laplacian-of-Gaussian filter from motion corrected- (using MotionCorr2) and CTF corrected-micrographs (using CTFFIND4.1). Multiple rounds of 2D class averaging were performed to remove junk particles, and an ab initio model was constructed from 720,372 particles. 3D classification with alignment yielded 3 major classes (Classes I -III). Refinement, CTF refinement, and Bayesian polishing

followed by postprocessing were performed yielding final 3.8 Å (class I), 4.3 Å (class II), and 4.3 Å (class III) resolution maps based on 0.143 FSC cutoff criteria. Structures of ATAD2 chains (obtained from the ATAD2 Walker B alone cryo-EM map) were fit into ATAD2-H3/H4 complex maps and real-space refined in Phenix[54]. Final cryo-EM maps and structure coordinates are deposited in EMDB (EMD-34468, 36665, 36666, and 36667) and PDB (ID: 8H3H, 8JUW, 8JUY, and 8JUZ).

**ATPase assays.** The EnzChek Phosphate assay kit (Thermo Fisher Scientific) was used to measure steady state ATPase rates of ATAD2. ATAD2 at a concentration of 200–300 nM was incubated in assay buffer (50 mM Tris (pH 8.0), 100 mM NaCl, 1 mM DTT) at 37 °C and 1 mM ATP was added to initiate reactions. Absorbance at 360 nm was monitored on a Tecan Spark microplate reader at 10 s intervals.

Other detailed methods are described in the Supplementary Methods.

**Reporting summary.** Further information on research design is available in the Nature Portfolio Reporting Summary linked to this article.

## Data availability

The cryo-EM maps and the final refined coordinates for the structures reported in this manuscript have been deposited to the EMDB and PDB, respectively, with the following accession codes: ATAD2 Walker B ATP (EMD-34468, 8H3H), ATAD2-H3/H4K5Q Class I (EMD-36665, 8JUW), ATAD2-H3/H4K5Q Class II (EMD-36666, 8JUY), and ATAD2-H3/H4K5Q Class III (EMD-36667, 8JUZ).

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

## Acknowledgements

We would like to thank Sarah Sterling and Richard Walsh at the Harvard Center for Cryo-Electron Microscopy (HC2EM), and Sujeong Kim at POSTECH Institute of Membrane Proteins (IMP) for supporting data collection. We also thank Yumi Shin and Eunhee Seong for technical assistance, and Dr. Bob Kingston for supporting this project. This work was supported by grants from the National Research Foundation of Korea (NRF). Specifically, a Sejong Science Fellowship (2022R1C1C2003419) and the Basic Science Research Program through the Korea Ministry of Education (2019R1A6A1A10073887) to C.C. and grants (2020R1A2B5B03001517, 2020M3E5E2037170, RS-2023-00266300) and the framework of international cooperation program (2021K2A9A2A08000088) to J.S. This research was also supported by Joint Research of the Exploratory Research Center on Life and Living Systems (ExCELLS program No. 18-101 to T.U., 22EXC601 to T.U. and K.K., 20-318, 21-313, and 22EXC305 to J.S) and the JSPS Bilateral Program (Grant number JPJSBP120218819) to K.K.

## Author contributions

C.C. conceived the study, designed experiments, purified proteins, performed biochemical and XL-MS experiments, analyzed cryo-EM structure, prepared figures, wrote manuscript, and provided funding. C.G. designed, performed and analyzed HS-AFM experiments. T.U. and K.K. guided HS-AFM experimental design and analysis, provided input on the manuscript, and provided funding. J.-J.S. conceived the study, performed cryo-EM experiments and analysis, wrote the manuscript, and provided funding.

## Competing interests

J.-J.S. is a co-founder and CTO of Epinogen. All other authors declare no competing interests.
