## [Peer Review File · Communications Biology]

Reviewers' comments:

Reviewer #1 (Remarks to the Author):

Comments:

Cho et al here present cryoEM structures of another ATAD2 protein from human in addition to their previous work on yeast ATAD2 homolog Abo1. The authors demonstrate cryoEM structures of ATAD2 in its ATP binding state and ATAD2-H3/H4 complex. An unusual autoinhibited state of ATAD2 was captured. Additionally, the binding of ATAD2 with H3/H4 complex was analysed in combination of structure obtained and XL-MS. A comprehensive comparison of ATAD2 and yeast Abo1 was done, unravel the possible function of ATAD2. Moreover, the author here discovered a novel gate loop of ATAD2 that function as ISS (inter-subunit signalling) corresponding to other AAA+ protein. Overall, the work shows interest, and the author did deep investigation by comparison the structures obtained with its homologs and clarify the new findings.

Although in general the work is acceptable, the authors do need to improve the expression, correct grammar mistakes and rephrase the words to make the manuscript more readable, avoid expression that cause confusions and make it easy to follow for the readers.

There are many places in the manuscript that need to rephrase:
Here list some examples:

Page 3 line "60 biological context, suggesting that even the transcriptional functions of ATAD2 are poorly 61 understood."

Page 4 line 72-73 "assembly assay19, suggesting that Abo1 acts as a histone chaperone like Yta7, but instead catalyzes nucleosome assembly instead of disassembly".

Page4 Line 82:"opposes the activity of the HIRA and FACT", it would be odd to use "oppose" here, I guess could be "inhibit" used here.

Page5 Line 95 : "As opposed to most other histone chaperones", here better use "in contrast", not "opposed"

Page 8 Line 161-163 "No density for bromodomains, linker arms, and C-terminal linker domains were visible", it indicates there is no density for those domains, which obvious here the expression is odd.

Additionally, there are some minor mistakes or information missing.

Page 3 line 46 "ATAD2 (ATPase family AAA+ domain-containing 2)", it is normally described as "ATPase family AAA domain-containing 2"

Page 5 111 "organization of an ATAD2 homolog, showing that Abo1 assumes a three-tiered hexamer", "Assume" is not the right word depicted here, might use "displays"

Page 7 Line 150 "ATAD2 Walker B mutant protein was purified by sequential affinity, ion exchange, and size exclusion chromatography, and the peak size exclusion fractions corresponding to hexameric size" It should be the "peak fractions of size exclusion".

Page 13 294-295 "where only 5 aa could be modelled, as opposed to 14 aa or 22 aa-long substrates in Abo1or translocating unfoldases such as p97". "As opposed to" should be "in contrast to"

Supplementary fig2, fig8., etc: scale bar in the figs to show the size of the particles as well as in 2D class average.

474 "950mL of SF9 suspension cultures at a density of 2.5-3.5 x 10⁶ cells with 50mL of P3 virus 475 for 45-48hrs", the expression makes readers confused, please rephrase.

Page 22 line 476 "purified by inclusion body purification as described in Luger et al 44". replace "in" with "by"

Page 23 line 498 "Histone H3/H4 complex was prepared as described in Luger et al 44". replace "in" with "by"

Page 24 Line 513-515 "added to proteins and incubated for 30min @ 37 degrees and quenched by the addition of"; "20min incubation @ 515 37 degrees." Why "@" used here.

Page 24 line 532-550: the citations are missing of those programs used for data processing and structure refinement, Like CTFFIND4.1, MotionCorr, COOT, PHENIX., etc.

Page 24 Line 525, Titan Krios 300 keV; Page 24 line 529 Talos Artica 200kEV, normally the unit is "kV".

Reviewer #2 (Remarks to the Author):

Summary

In this manuscript Cho and Song present three cryo-EM structures of the N-terminally truncated human ATAD2 (aa 403-1390), an ATP-dependent histone chaperone, one in absence and two presence of the histone substrate H3/H4. All structures harbour a Walker B mutation, which allows for only very low levels of ATP hydrolysis. The ATAD2 structure without the H3/H4 substrate reveals a shallow spiral arrangement and an unspecified 5-amino acid peptide bound in its central pore. A similar, but more pronounced spiral arrangement has been previously described in other AAA+ ATPases, such as NSF and p97, and the author's own work on the ATAD2 *S. pombe* homolog, Abo1. In case of ATAD2, the spiral arrangement is more planar, and the usual seam where the ADP-bound subunit meets the adjacent, ATP-bound subunit is tighter than usual, resulting in less disorder in seam subunits. In presence of nucleotide, the tight arrangement is aided by a conserved N-terminal linker domain that is tightly packed between the seam subunits. General instability of the hexamer in absence of nucleotide compared to the *S. pombe* homolog is explained by lack of a knob-hole structure, which helps to stabilize the inter-subunit interactions in its yeast counterpart. In addition to that, a gate loop that mediates inter-subunit communication has been identified, that could lead to a new direction in ATAD2 inhibitor developments. Upon incubation with the target histone substrate H3/H4 no drastic conformational changes occur in the spiral arrangement of ATAD2, however, an increased disorder in the N-terminal linker domain is observed, supported by XL-MS. This could be a consequence of allosteric changes that occur upon H3/H4 binding, however, the exact sequence of events and the mechanism behind it remains unclear.

Overall impressions

The manuscript is conceptually well written and the structural data for the given protein fragment used is overall convincing (more for ATAD2 without, than with the H3/H4 substrate), although some points are to be further elaborated on or depicted in a clearer way. The comparisons of ATAD2 without the H3/H4 substrate with the yeast homologs and across the human structures are insightful and comprehensively written, so it is easy to follow what the main similarities and differences are compared to these proteins and other AAA+ ATPases. As a highlight of the paper, I would point out the newly identified auto-regulatory mechanism of ATAD2, which sets it apart from other examples in this protein family. Also, mapping of the gate loops as the ISS motif equivalent is an interesting finding, with exciting implications for drug development. I find this part of the results well thought out and nicely presented.

A weak point of this work is that the authors almost exclusively rely on structural data and provide little biochemical data to validate the conclusions drawn from structural analysis, even when specific interactions have been determined and functional data would complete the picture. Additionally, I find that not all claims are supported by data and figures, especially related to the allosteric regulation of ATAD2 by the H3/H4 substrate. This part of the manuscript seems least convincing and lacking in detail, although it has potential to answer those most exciting questions related to substrate engagement. Positive side is that XL-MS was performed to help pinpoint more details on the interaction between ATAD2 and its substrate. In terms of writing, final polishing steps are required, and I would recommend proofreading the final version before submitting in the future. In conclusion, I do consider this work to be of interest for scientists working on ATAD2 and AAA+ ATPases, however, there is room for improvement to get the manuscript to its final version and I hope you take the suggestions below to do so.

Points to consider:

1. Although the authors have shown that the yeast homolog Abo1 does not need the N-terminal domain for its function and stability, I do not understand why this has been extended to the human homolog. Was there a specific reason why the full-length ATAD2 couldn't be expressed and purified? What has been tried? I see you touch upon this in the Discussion part at the end of the manuscript, but this kind of information is also valuable for the community and could be included earlier in the manuscript.
2. In Supplementary Fig. 1a the "WT apo" sample trace in grey does not run with a straight baseline and therefore cannot be used for comparison purposes. Please correct for the incline or repeat the run.
3. In Supplementary Fig. 1b please increase the micrograph size and mark examples of hexamer vs. other oligomerization states. Also, very different protein concentrations seem to have been used for each protein variant, which makes the comparison by visual inspection challenging. Better statistics could be obtained from 2D classification, however if you correct the "WT apo" sample trace in grey in Supplementary Fig.1a, that would be sufficient.
4. Why is there no measurable ATPase activity of the wild-type ATAD2? Is it due to using a truncated construct? Does it show activity in presence of the histone substrate? Including the ATP-ase activity measurements would be very informative (also for the LD mutants mentioned later).
5. The packing of the HBD with AAA1 in ATAD2 vs. AAA2 in Abo1 is very interesting. When looking at the interacting surface residues, is the level of sequence conservation high enough to allow for further shifting of the HBD in ATAD2 down towards the AAA2 domain? Do you think this could happen in wild type? In Fig. 2d and 2e please mark the HBD in a different colour or make it lighter/darker to make it distinguishable from the overall P6 subunit. This way you can nicely emphasize your point from the main text (lines 185-195).
6. When considering the rigid-body fit of the two bromodomains in a head-to-tail arrangement (line 324 on) I do not find it convincing to compare it to a structure that has also used a rigid body fit (at

least to fit the second bromodomain). I guess you are referring to PDB ID 7UQI / emd-26695 in your comparison (please provide this information). Can you find other, high-resolution structures to support this bromodomain assembly? Please show a closeup of the bromodomain fit. Also please update Citation 41, as this work has been published in JBC in the meantime.

7. Structural analysis of the ATAD2 structure with the H3/H4 substrate bound is somewhat underwhelming. How much does the ring open compared to the structure without H3/H4? What is the nucleotide occupancy? What is the mechanism of the allosteric regulation? Can you propose a sequence of events? There is no figure to highlight differences compared to the unbound ATAD2. Also, there seems to be an error in line 316 ("the N-terminal tails and histone bodies of H3 and H3...").

8. What is the impact of using the Walker B mutants when looking at the interaction with the substrate? Can you share your thoughts on that? Clearly, using those mutants helps to reduce sample heterogeneity and improves hexamer stability, but it would be valuable to include your comment in the manuscript. For example, do you think that the AAA2 would remain an "inert base" upon substrate engagement in the wild-type protein? Or would it contribute in some form of interaction or substrate processing?

9. In Supplementary Fig. 3b there seems to be a mismatch in the local resolution colour scale showing regions resolved up to 2.5 Å and resolution obtained as described in your manuscript for the ATAD2 Walker B mutant (according to your FSC 3.15 Å). Please show a relevant resolution scale starting with the highest obtained resolution and ending with a suitable resolution for the poorly resolved map edges. Similar point is valid for Supplementary Fig. 9.

10. In Methods, please include more details and specify the tools used for de novo model building (line 539), as this is an uncommon procedure for model building when structural homologs are available. If you have used a homology model, refer to the modelling server and/or template structure used.

Minor points:

11. For clarity I suggest using a term "N-terminally truncated human ATAD2" rather than "near-full-length human ATAD2" (Introduction, line 120), since ca 400 out of 1400 amino acids have been excluded.

12. EM maps with resolution > 3.0 Å are usually not considered high resolution in the field. I suggest using "sufficiently high resolution" or excluding this claim (line 201-202).

13. In Fig. 2b the nucleotide EM densities are very pale, and the images (including the used font size) are small. I suggest making the EM density a bit darker and showing the image in full page width, since it is a nice depiction and an important point to show when claiming certain nucleotide occupancy.

14. In Fig. 5a (right) would be good to see the EM density for the substrate chain in the same style as shown for the nucleotides in Fig. 2b (after you adjust that one).

15. Please refrain from using the term "atomic resolution" (line 362). This is a commonly misused and undefined term, maybe suitable for EM maps that are breaking the resolution limits in the field (<1.5 Å), but it is not applicable for EM maps with resolution >3.5 Å, where one cannot resolve individual atoms.

16. Citation missing for comparison with Msp1 (lines 220-223).

17. Please proofread the Methods part and correct for chemical names and missing or non-standard units. Some examples:

- in all instances the pH of buffers is written as "(8.0)" – please change to "(pH 8.0)"

- "β-mercaptoethanol" (line 503)

- "20min incubation @ 37 degrees" (line 514)

- in almost all instances space is lacking between number and unit ("50mL" instead of "50 mL")

- in all instances "hrs" is used a time unit – please use "h" or "hours"

- etc.

18. Please proofread and improve Supplementary Information, as some of the figure descriptions are written in an informal way (for example Supplementary Fig. 8 "+ ATP" could be written as "in

presence of ATP" and the H3/H4 terminology should be used in the same way throughout the paper, instead of "H3H4").

Reviewer #3 (Remarks to the Author):

In the manuscript by Cho et al, the authors determined the structure of the ATAD2, a member of the AAA+ ATPase family that also contains a bromodomain. ATAD2 has been identified as a transcriptional regulatory protein that is highly overexpressed in several cancers. However, the molecular function of this protein has been elusive due to the inability to study it directly in vivo. Cho et al, were able to successfully express and purify an N-terminal truncated version of the ATAD2 protein, starting at the sequence just before the first AAA domain and ending after the C-terminal domain. While understanding the structural organization of this protein is of high interest the novelty of the structural information as presented here is rather low. The authors have already published a structure of the ATAD2 homolog from yeast, Abo1, which has a very similar organizational structure and fold. The Abo1 manuscript published by Cho et al elegantly connected the structure of Abo1 to its function via ATPase activity assays, histone loading assays, and HS-AFM to show that the ATPase activity was required to place histone H3-H4 onto naked DNA and that one of the 6 subunits was involved in the catalytic activity driving the molecular motor functions, and had a conformational change as ATP was utilized by this molecular motor. This established Abo1 as a chromatin remodeler involved in histone loading. Another homolog of ATAD2, Yta7 has since been shown to remove histones from the nucleosome. However, the function of human ATAD2 is currently unknown, and the current manuscript does little more than present the structure of ATAD2 as a hexameric complex. Beyond that the claims by the authors about the function of ATAD2 are not supported by the data presented. The MS data shown does indicate that ATAD2 interacts with the H3-H4 histones as one would expect, but the structural data presented to support the configuration of the bromodomain region and the histones is not of high enough quality/resolution to make any conclusions, and it is not clear that the regions identified represent the bromodomain or histone molecules since the sequence of the protein in these regions could not be determined. Thus, the descriptions of these interactions are highly speculative. Overall, there are several major flaws in the data presented that make this study unsuitable for publication.

1. The methods sections are incomplete at best, and in many cases completely absent. This raises questions about if the approach used, and if any artifacts were introduced that contribute to non-biological interactions. For example, there no methods presented for any of the data in the supplementary materials. This includes SEC chromatography (what column), how were molecular weights determined, for the negative stain images, what stain was used, and what protein concentrations, for the MS data there is NOTHING written to evaluate how the experiment was performed. The rigor and reproducibility of the experiments presented is not existent and not acceptable.

2. The lack of any ATPase activity in the ATAD2 protein construct they obtained is highly concerning. It appears the authors have not purified an active complex so much of the discussion that related to conformational changes during catalysis are not valid. As such, the authors have not determined if ATAD2 is a chaperone or chromatin remodeler and any reference to that need to be removed from the discussion. Furthermore, although this is an impressive structure at 3.1Å any regions of the structure that analyzes conformational changes, particularly hydrogen bond interactions, intersubunit contacts, and gate loop opening/closing need to be supported by density of that specific region. There is only one figure in the supp data that shows the quality of the local resolution and it isn't even stated what part of the structure/density is shown. The resolution of structures determined by CryoEM is highly variable from region to region. For example, the AAA1 ATPase domain has ~3Å resolution, but the bromodomain region has greater than 12Å resolution. It is not possible to describe bond interactions

at such low resolution where secondary structure may not be discernable.

3. The figures are poorly labeled and confusing to follow. The descriptions and labeling need to be improved. For example:

- Figure 1B should have scale bars for the 2D classes. The model in Fig 1e, the linker arm regions (dotted lines) don't accurately represent the amino acid lengths. Also, the bromodomain in this figure should be greyed out, there is no evidence in structures presented in figure one for where the bromodomain is located with respect to the AAA domains.
- Figure 2 D, E, f, g more labelling is needed. It is not clear which structures are ATAD2 vs Abo1. Abbreviations should be spelled out in the figure legend. Color coding for the first and second AAA domains would clarify D-E, or labeled. Things that are discussed in the manuscript text are not labeled in D-E, or other parts of this figure. Figure 2 would be improved by having top side and bottom views.
- Fig 3b no information is given for the sigma value of the density map. Also for the conformational changes to be supported the density needs to be shown for the gate loop region. Figures c, d and E are very difficult to follow or understand what is being presented. The color coding between subunits is not matching between the panels. P6 is pink in A and E, and Red and B, C, D.
- Figure 4 need to show the density to support conformation and bond interactions of the LD region. Also a comparison of the density between other subunits would be helpful to compare the difference between the seams of P1-P6 vs P2-P3 for example.
- Figure 5, is difficult to follow and not labeled well, particularly C and D. The comparison between Abo1 and ATAD2 show much different views of the structure in A and B. The insert of the model in A, should include density around the model.
- Figure 6. The density supporting placement of the bromodomain and histone is low resolution or poor quality such that no sequence information can be derived and it is unclear that the density represents these proteins/regions. The MS data to support the interactions has no method information, and it is quite possible that the crosslinking performed to establish the ATAD2-histone complex created non-natural configurations, or are not the proteins expected. The bromodomain model does not look like it fits well into their density, particularly bromodomain 2. More detailed figures in the supp data may help.

In summary, while the cryoEM structure of ATAD2 is important to study many of the claims made in the manuscript are unfounded based on the data presented. This is particularly true for the function of LD domain, the ATAD2-histone interactions, and the position of the bromodomain. The lack of detailed and missing methods is extremely concerning. As such, the study is not acceptable for publication.

Additional specific comments:

- In the line 334 authors have found ATAD2 interaction with histone H3/H4 dimer but from the material method section line 505 it looks like they used histone H3/H4 tetramer
- It will be helpful for the researchers in the field to mention in the discussion section about the difficulties/reasons of not obtaining a good resolution for ATAD2/H3-H4 complex structure compared to ATAD2 alone.
- Line 322-325 states in the class 1 (ATAD2 alone at 4.7 Å) authors have found the extra density for bromodomain. When they tried a cryo-EM study with ATAD2 + ATP alone they achieved a better resolution of 3.1 Å but no density for bromodomain was observed. State any specific reason in the discussion section of not obtaining a density for bromodomain at 3.1 Å structure.
- Line 307-311 states that AAA1, bromo and C-terminal linker domain of ATAD2 interacts with histone H3/H4 and Line 334 states ATAD2 binds to one H3/H4 dimer. ATAD2 being hexamer then why it accommodates only one H3/H4 dimer why not six? I would suggest including this point in the discussion section.
- I would also recommend doing some additional biophysical experiment such AUC or SAXS to conform the ratio and binding of ATAD2/H3-H4.
- Line 509-512 states that ATAD2 and H3/H4 was incubated at 150 mM NaCl but histone H3/H4

tetramer was purified at 2M NaCl. Does lower of the salt affect the stability of histone H3/H4 tetramer if not then how the salt conc. was decreased for the binding reaction can be written in detail for replicating the result by other researchers in the field.

- Line 515-517 states that author have used sucrose gradient for the purification of ATAD2/H3-H4 crosslinked sample. I would suggest adding the SDS-PAGE gel image in the of the fractions obtained from the sucrose gradient as supplementary figures for better clarity.

We thank the reviewers for many helpful suggestions and comments. According to the reviewers' suggestions, we have revised the figures and text (please refer to point-by-point discussions below) which we hope improve the clarity and readability of the manuscript. We have also added more information regarding the methods and provided detailed maps of various regions of our cryo-EM structure (new Supplementary fig. 4) that should help validate our ideas on AAA gate loop conformation and N-term LD based auto-inhibition.

In order to address major concerns about 1) the low resolution of the ATAD2-H3/H4 structure, and 2) the validity of using a catalytically inactive Walker B mutant as a basis of mechanistic interpretation, we have added two pieces of additional data. First, we have improved the resolution of the ATAD2-H3/H4 structure (from 4.6 Å to 3.8 Å) by collecting a larger cryo-EM dataset with a 300kV microscope using H4K5ac mimic histone H3/H4 crosslinked to ATAD2. This improved structure shows a clearer extra density that connects to the AAA+ ring pore and supports the idea that the extra density observed is part of the H3/H4 substrate. We also find a previously unobserved conformation where the AAA+ ring has significant asymmetry with a completely disordered subunit and a broken seam supporting our previous idea that H3/H4 binding induces symmetry breaking and release of the autoinhibitory N-terminal LD from the seam. Second, we have obtained HS-AFM movies of ATAD2 Walker B mutant dynamics in the absence and presence of H3/H4 (Supplementary fig. 14). These movies show that even though the ATAD2 Walker B mutant has no measurable bulk ATPase activity, dynamic symmetry breaking still does occur suggesting that the ATAD2 Walker B mutant conformation retains active AAA+ ATPase-like properties. Furthermore, the frequency of symmetry breaking increases in the presence of histone H3/H4, consistent with the idea that H3/H4 might allosterically prime ATAD2. We hope that these improvements help bolster our claims about ATAD2 activation, and generally improve the quality of the manuscript.

Reviewers' comments:

Reviewer #1 (Remarks to the Author):

Comments:

Cho et al here present cryoEM structures of another ATAD2 protein from human in addition to their previous work on yeast ATAD2 homolog Abo1. The authors demonstrate cryoEM structures of ATAD2 in its ATP binding state and ATAD2-H3/H4 complex. An unusual autoinhibited state of ATAD2 was captured. Additionally, the binding of ATAD2 with H3/H4 complex was analysed in combination of structure obtained and XL-MS. A comprehensive comparison of ATAD2 and yeast Abo1 was done, unravel the possible function of ATAD2. Moreover, the author here discovered a novel gate loop of ATAD2 that function as ISS (inter-subunit signalling) corresponding to other AAA+ protein. Overall, the work shows interest, and the author did deep investigation by comparison the structures obtained with its homologs and clarify the new findings.

Although in general the work is acceptable, the authors do need to improve the expression, correct grammar mistakes and rephrase the words to make the manuscript more readable, avoid expression that cause confusions and make it easy to follow for the readers.

> We thank reviewer #1 for their comments. We have made the suggested changes and substantially revised the figures (please see reply to Reviewers #2 and #3) to improve clarity and readability. We hope that these changes have improved the manuscript and made it more accessible.

There are many places in the manuscript that need to rephrase:

> We have rephrased the following lines to improve clarity.

Here list some examples:

Page 3 line "60 biological context, suggesting that even the transcriptional functions of ATAD2 are poorly understood."

Page 4 line 72-73 "assembly assay¹⁹, suggesting that Abo1 acts as a histone chaperone like Yta7, but instead catalyzes nucleosome assembly instead of disassembly".

Page 4 Line 82: "opposes the activity of the HIRA and FACT", it would be odd to use "oppose" here, I guess could be "inhibit" used here.

Page 5 Line 95 : "As opposed to most other histone chaperones", here better use "in contrast", not "opposed"

Page 8 Line 161-163 "No density for bromodomains, linker arms, and C-terminal linker domains were visible", it indicates there is no density for those domains, which obvious here the expression is odd.

Additionally, there are some minor mistakes or information missing.

> We have revised all of the following to correct for missing or incorrect information.

Page 3 line 46 "ATAD2 (ATPase family AAA+ domain-containing 2)", it is normally described as "ATPase family AAA domain-containing 2"

Page 5 111 "organization of an ATAD2 homolog, showing that Abo1 assumes a three-tiered hexamer", "Assume" is not the right word depicted here, might use "displays"

Page 7 Line 150 "ATAD2 Walker B mutant protein was purified by sequential affinity, ion exchange, and size exclusion chromatography, and the peak size exclusion fractions corresponding to hexameric size" It should be the "peak fractions of size exclusion".

Page 13 294-295 "where only 5 aa could be modelled, as opposed to 14 aa or 22 aa-long substrates in Abo1 or translocating unfoldases such as p97". "As opposed to" should be "in contrast to"

Supplementary fig2, fig8., etc: scale bar in the figs to show the size of the particles as well as in 2D class average.

Page line 474 "950mL of SF9 suspension cultures at a density of 2.5-3.5 x 10⁶ cells with 50mL of P3 virus 475 for 45-48hrs", the expression makes readers confused, please rephrase.

Page 22 line 476 "purified by inclusion body purification as described in Luger et al 44". replace "in" with "by"

Page 23 line 498 "Histone H3/H4 complex was prepared as described in Luger et al 44". replace "in" with "by"

Page 24 Line 513-515 "added to proteins and incubated for 30min @ 37 degrees and quenched by the addition of"; "20min incubation @ 37 degrees." Why "@" used here.

Page 24 line 532-550: the citations are missing of those programs used for data processing and structure refinement, Like CTFIND4.1, MotionCorr, COOT, PHENIX., etc.

Page 24 Line 525, Titan Krios 300 keV; Page 24 line 529 Talos Artica 200keV, normally the unit is "kV".

Reviewer #2 (Remarks to the Author):

Summary

In this manuscript Cho and Song present three cryo-EM structures of the N-terminally truncated human ATAD2 (aa 403-1390), an ATP-dependent histone chaperone, one in absence and two presence of the histone substrate H3/H4. All structures harbour a Walker B mutation, which allows for only very low levels of ATP hydrolysis. The ATAD2 structure without the H3/H4 substrate reveals a shallow spiral arrangement and an unspecified 5-amino acid peptide bound in its central pore. A similar, but more pronounced spiral arrangement has been previously described in other AAA+ ATPases, such as NSF and p97, and the author's own work on the ATAD2 *S. pombe* homolog, Abo1. In case of ATAD2, the spiral arrangement is more planar, and the usual seam where the ADP-bound subunit meets the adjacent, ATP-bound subunit is tighter than usual, resulting in less disorder in seam subunits. In presence of nucleotide, the tight arrangement is aided by a conserved N-terminal linker domain that is tightly packed between the seam subunits. General instability of the hexamer in absence of nucleotide compared to the *S. pombe* homolog is explained by lack of a knob-hole structure, which helps to stabilize the inter-subunit interactions in its yeast counterpart. In addition to that, a gate loop that mediates inter-subunit communication has been identified, that could lead to a new direction in ATAD2 inhibitor developments. Upon incubation with the target histone substrate H3/H4 no drastic conformational changes occur in the spiral arrangement of ATAD2, however, an increased disorder in the N-terminal linker domain is observed, supported by XL-MS. This could be a consequence of allosteric changes that occur upon H3/H4 binding, however, the exact sequence of events and the mechanism behind it remains unclear.

Overall impressions

The manuscript is conceptually well written and the structural data for the given protein fragment used is overall convincing (more for ATAD2 without, than with the H3/H4 substrate), although some points are to be further elaborated on or depicted in a clearer way. The comparisons of ATAD2 without the H3/H4 substrate with the yeast homologs and across the human structures are insightful and comprehensively written, so it is easy to follow what the main similarities and differences are compared to these proteins and other AAA+ ATPases. As a highlight of the paper, I would point out the newly identified auto-regulatory mechanism of ATAD2, which sets it apart from other examples in this protein family. Also, mapping of the gate loops as the ISS motif equivalent is an interesting finding, with exciting implications for drug development. I find this part of the results well thought out and nicely presented.

A weak point of this work is that the authors almost exclusively rely on structural data and provide little biochemical data to validate the conclusions drawn from structural analysis, even when specific interactions have been determined and functional data would complete the picture. Additionally, I find that not all claims are supported by data and figures, especially related to the allosteric regulation of ATAD2 by the H3/H4 substrate. This part of the manuscript seems least convincing and lacking in detail, although it has potential to answer those most exciting questions related to substrate engagement. Positive side is that XL-MS was performed to help pinpoint more details on the interaction between ATAD2 and its substrate. In terms of writing, final polishing steps are required, and I would recommend proofreading the final version before submitting in the future. In conclusion, I do consider this work to be of interest for scientists working on ATAD2 and AAA+ ATPases, however, there is room for improvement to get the manuscript to its final version and I hope you take the suggestions below to do so.

> We thank Reviewer #2 for the detailed and thorough critique of our work. We agree that additional analyses and clarifications would strengthen the arguments and improve the overall quality of the manuscript, and have taken steps to address these issues. Please refer to the point-by-point discussions below:

Points to consider:

1. Although the authors have shown that the yeast homolog Abo1 does not need the N-terminal domain for its function and stability, I do not understand why this has been extended to the human homolog. Was there a specific reason why the full-length ATAD2 couldn't be expressed and purified? What has been tried? I see you touch upon this in the Discussion part at the end of the manuscript, but this kind of information is also valuable for the community and could be included earlier in the manuscript.

> In our initial attempts to purify both full length wtATAD2 and Walker B mutant ATAD2 protein, we encountered challenges including low yields, proteolysis, and severe aggregation under various conditions. This may be attributed to the nature of the ATAD2 N-terminus, which is predicted to be unstructured and prone to phosphorylation and ubiquitination. However, by removing the N-terminus similar to the testis-specific short isoform of ATAD2 (ATAD2-S, Caron et al, 2010) we were able to successfully purify high yields of protein without aggregation and proteolysis.

Thus, for the purpose of our structural study, we chose to use the truncated form of ATAD2 for its biochemical tractability and similarities to the testes-specific isoform and the functional fission yeast ATAD2 (Abo1) homolog. Nonetheless, it is important to note that the N-terminus might play a crucial role in regulating ATAD2 activity, as suggested for budding yeast ATAD2 (Chacin et al, 2021). In future studies, we will attempt to purify and characterize the full-length version of ATAD2.

> We have accordingly added the comments to the Results and Discussion section of the manuscript:

-pg7 line 141; "In our initial attempts to purify full length ATAD2, we encountered challenges including low protein yield, proteolysis, and severe aggregation under various conditions. However, by removing the acidic N-terminal domain to create a construct similar to the testis-specific short isoform of ATAD2 isoform (ATAD2-S)¹, we were able to increase protein yields and homogeneity. Thus, in this study, we used N-terminally truncated ATAD2 (aa 403-1390, Fig. 1a) in parallel to our previous studies of Abo1..."

-pg24 line 557; "a limitation of this study is the inability to obtain a catalytically active form of full-length wild type ATAD2. The truncated form of wild type ATAD2 used in this study lacks the ability to form and maintain stable hexamers (Supplementary fig. 1) with active nucleotide binding pockets, which might be due to the absence of specific post-translational modifications necessary for ATAD2 activation, or the exclusion of the ATAD2 N-terminus in our constructs. Supporting these ideas, ATAD2 has been shown to undergo extensive phosphorylation and ubiquitylation at multiple sites, and in the case of Yta7, it has been shown that N-terminal phosphorylation is essential for catalytic activation. Alternatively, an additional protein factor might be required to stabilize and activate the functional hexameric form of ATAD2."

2. In Supplementary Fig. 1a the "WT apo" sample trace in grey does not run with a straight baseline and therefore cannot be used for comparison purposes. Please correct for the incline or repeat the run.

> We have repeated the run to obtain a straight baseline, and accordingly updated **Supplementary fig. 1**.

3. In Supplementary Fig. 1b please increase the micrograph size and mark examples of hexamer vs. other oligomerization states. Also, very different protein concentrations seem to have been used for

each protein variant, which makes the comparison by visual inspection challenging. Better statistics could be obtained from 2D classification, however if you correct the “WT apo” sample trace in grey in Supplementary Fig.1a, that would be sufficient.

> In order to better enable comparison of different ATAD2 proteins, we have added a native gel in **Supplementary fig. 1b**, and presented enlarged micrographs that have similar particles densities in **Supplementary fig. 1c**. We have also added representative 2D classes of each protein to illustrate the heterogeneity of wt and N-terminal LD mutant ATAD2 compared to Walker B mutant ATAD2.

4. Why is there no measurable ATPase activity of the wild-type ATAD2? Is it due to using a truncated construct? Does it show activity in presence of the histone substrate? Including the ATPase activity measurements would be very informative (also for the LD mutants mentioned later).

> Based on our ATPase activity assays, the ATPase rate of wild type ATAD2 is < 0.03 ATP/min/hexamer, which is $\sim 3,000$ times slower than that of Abo1 (90 ATP/min/hexamer), and close to the detection limits of our ATPase assay. The ATPase rate of Walker B mutant ATAD2 is slightly lower at 0.01 ATP/min/hexamer, and the rates are unchanged by the addition of histone substrate.

We did not include ATPase data as we were unable to observe robust ATPase activity, as expected on the level for an active cycling AAA+ ATPase. We are uncertain why the wild type protein does not have ATPase activity, but it likely relates to the heterogeneous conformations of ATAD2 we observe by gel filtration and negative EM.

> We have accordingly added related comments on **pg. 7 line 153**: “... the catalytic ATPase activity of wild type ATAD2 was measured to be 0.03ATP/hexamer/min, a rate approximately 3,000 times slower than Abo1 and close to the detection limit of our ATPase assays, implying that truncated ATAD2 did not maintain stable hexamers with functional nucleotide pockets.”

5. The packing of the HBD with AAA1 in ATAD2 vs. AAA2 in Abo1 is very interesting. When looking at the interacting surface residues, is the level of sequence conservation high enough to allow for further shifting of the HBD in ATAD2 down towards the AAA2 domain? Do you think this could happen in wild type? In Fig. 2d and 2e please mark the HBD in a different colour or make it lighter/darker to make it distinguishable from the overall P6 subunit. This way you can nicely emphasize your point from the main text (lines 185-195).

> We thank the reviewer for the suggestion, and have highlighted the AAA1 HBD in lighter hues in **Figs. 2d and e** to help differentiate between the NBD and HBD. The sequence conservation between the ATAD2 and Abo1 AAA1 domains is $\sim 90\%$, but the conservation of the AAA2 domains is quite low at $\sim 35\%$. Also, the sequence conservation between the AAA1 and AAA2 domains in the same protein is even lower at 25%, so it is difficult to draw any conclusions on the plausibility of the ATAD2 HBD domain moving further down towards the AAA2 domain based on sequence conservation alone. However, based on the asymmetric conformation we observe in our new HS-AFM experiments, and the improved cryo-EM structure of the ATAD2-H3/H4K5Q complex, we do think that the AAA1 HBD of ATAD2 could shift down towards the AAA2 NBD during the wild type cycle, as mentioned in our discussion.

6. When considering the rigid-body fit of the two bromodomains in a head-to-tail arrangement (line 324 on) I do not find it convincing to compare it to a structure that has also used a rigid body fit (at least to fit the second bromodomain). I guess you are referring to PDB ID 7UQI / emd-26695 in your comparison (please provide this information). Can you find other, high-resolution structures to support this bromodomain assembly? Please show a closeup of the bromodomain fit. Also please update Citation 41, as this work has been published in JBC in the meantime.

> We have updated citation 41 to reflect the publication of the work.
> When analyzing our new cryo-EM data, we again found the handle-like extra density on the top surface of the AAA+ ring, but were unable to improve the resolution to higher resolutions. We agree that at such low resolution, it is premature to discuss bromodomain identity and conformation. As such, we have removed claims regarding the conformation and identity of the bromodomain and have moved the cryo-EM map and extra density of this class to Supplementary fig. 12.

7. Structural analysis of the ATAD2 structure with the H3/H4 substrate bound is somewhat underwhelming. How much does the ring open compared to the structure without H3/H4? What is the nucleotide occupancy? What is the mechanism of the allosteric regulation? Can you propose a sequence of events? There is no figure to highlight differences compared to the unbound ATAD2. Also, there seems to be an error in line 316 (“the N-terminal tails and histone bodies of H3 and H3...”).

> With the improved resolution of the ATAD2-H3/H4K5Q complex, we have been able to perform a more thorough analyses of changes in the AAA+ domains compared to ATAD2 alone (description in results **pg. 16-18, lines 381-412**). Overall, the most prominent differences are at the seam, where subunit P1 and P6 become significantly disordered. These changes are accompanied by a disappearance of the N-LD at the seam and movement of the P1 and P6 subunits further away from each other, and are illustrated in our updated **Supplementary Movies 2-4**, as well as in **new Fig. 6f-h**. The nucleotide states and gate loop conformations are similar in the presence and absence of H3/H4, although in the ATAD2-H3/H4K5Q complex Class II conformation, there is some ambiguity in the nucleotide state at the P1/6 and P2/1 interface due to weak nucleotide density.

> Based on H3/H4-dependent changes in the AAA+ ring as observed by cryo-EM, HS-AFM, and XL-MS, we propose that H3/H4 induces AAA+ ring symmetry breaking. As the H3/H4 tail does not seem to affect symmetry breaking in HS-AFM experiments, we propose that binding of the histone H3/H4 body to the bromodomains might allosterically activate the AAA+ domains by undocking of the N-terminal LD.

> We have corrected for the error in line 316.

8. What is the impact of using the Walker B mutants when looking at the interaction with the substrate? Can you share your thoughts on that? Clearly, using those mutants helps to reduce sample heterogeneity and improves hexamer stability, but it would be valuable to include your comment in the manuscript. For example, do you think that the AAA2 would remain an “inert base” upon substrate engagement in the wild-type protein? Or would it contribute in some form of interaction or substrate processing?

> Although the Walker B mutant used in our structural studies does not have wild type catalytic activity, we believe that the structure of the Walker B mutant likely reflects the structural principles of wild type ATAD2 (Please also refer to response to Reviewer #3 comments #2). We also think that the histone binding mode of H3/H4 on the top surface of the AAA+ ring would be conserved in wild type ATAD2, with the bottom AAA2 surface of the ring acting more as an “inert base”. We have included related comments in the Discussion **pg.24-25, lines 568-573**.

9. In Supplementary Fig. 3b there seems to be a mismatch in the local resolution colour scale showing regions resolved up to 2.5 Å and resolution obtained as described in your manuscript for the ATAD2 Walker B mutant (according to your FSC 3.15 Å). Please show a relevant resolution scale starting with the highest obtained resolution and ending with a suitable resolution for the poorly resolved map edges. Similar point is valid for Supplementary Fig. 9.

> The reported resolution of 3.15 Å (obtained from the FSC curve cutoff at 0.143) is a parameter that assesses the average resolution of the entire reconstructed volume. However, in practice, there are

substantial local variations in the resolution of the final reconstructed volume, and there are regions that have higher resolutions than the reported average value. Thus, tools that calculate local resolutions such as ResMap will output resolutions that are higher than the single reported value, and this is what is shown in the supplementary figures.

10. In Methods, please include more details and specify the tools used for de novo model building (line 539), as this is an uncommon procedure for model building when structural homologs are available. If you have used a homology model, refer to the modelling server and/or template structure used.

> We acknowledge that we may have used the term “de novo” loosely, as we initially used the structure of Abo1 to place the backbone, and subsequently mutated residues and refined side chains to build the model of ATAD2 residue by residue. To reflect this fact, we have removed any referrals of a “de novo” model, and provided more detail in the Methods (pg. 29, lines 662-665) as follows:

“Using the atomic model of Abo1 (PDB ID: 6JQ0) as an initial guide for backbone placement, residues of ATAD2 were built into the cryo-EM map using Coot followed by real space refinement in Phenix.”

Minor points:

11. For clarity I suggest using a term “N-terminally truncated human ATAD2” rather than “near-full-length human ATAD2” (Introduction, line 120), since ca 400 out of 1400 amino acids have been excluded.

> We have changed “near-full length ATAD2” to “N-terminally truncated ATAD2” as suggested.

12. EM maps with resolution > 3.0 Å are usually not considered high resolution in the field. I suggest using “sufficiently high resolution” or excluding this claim (line 201-202).

> We have excluded the claim of “high resolution” as suggested.

13. In Fig. 2b the nucleotide EM densities are very pale, and the images (including the used font size) are small. I suggest making the EM density a bit darker and showing the image in full page width, since it is a nice depiction and an important point to show when claiming certain nucleotide occupancy.

> We have enlarged Fig. 3b and made the nucleotide EM densities more prominent as suggested. Upon suggestions from Reviewer #3, we have also re-arranged panels in Fig. 3 and included more labels for all panels.

14. In Fig.5a (right) would be good to see the EM density for the substrate chain in the same style as shown for the nucleotides in Fig. 2b (after you adjust that one).

> We have added the EM density for the substrate in **Supp. Fig 4c**.

15. Please refrain from using the term “atomic resolution” (line 362). This is a commonly misused and undefined term, maybe suitable for EM maps that are breaking the resolution limits in the field (<1.5 Å), but it is not applicable for EM maps with resolution >3.5 Å, where one cannot resolve individual atoms.

> We agree with the comments, and have removed any referral to our structure as “atomic resolution” or “high resolution” considering that the resolution limit of cryo-EM has been pushed to “true atomic resolution”.

16. Citation missing for comparison with Msp1 (lines 220-223).

> We have added the citation.

17. Please proofread the Methods part and correct for chemical names and missing or non-standard units. Some examples:

in all instances the pH of buffers is written as “(8.0)” – please change to “(pH 8.0)”

“β-mercaptoethanol ” (line 503)

“20min incubation @ 37 degrees” (line 514)

in almost all instances space is lacking between number and unit (“50mL” instead of “50 mL”)

in all instances “hrs” is used a time unit – please use “h” or “hours”

- etc.

> We have corrected the errors as suggested.

18. Please proofread and improve Supplementary Information, as some of the figure descriptions are written in an informal way (for example Supplementary Fig. 8 “+ ATP” could be written as “in presence of ATP” and the H3/H4 terminology should be used in the same way throughout the paper, instead of “H3H4”).

> We have corrected for errors and used consistent terminology in the Supplementary Information.

Reviewer #3 (Remarks to the Author):

In the manuscript by Cho et al, the authors determined the structure of the ATAD2, a member of the AAA+ ATPase family that also contains a bromodomain. ATAD2 has been identified as a transcriptional regulatory protein that is highly overexpressed in several cancers. However, the molecular function of this protein has been elusive due to the inability to study it directly in vivo. Cho et al, were able to successfully express and purify an N-terminal truncated version of the ATAD2 protien, starting at the sequence just before the first AAA domain and ending after the C-terminal domain. While understanding the structural organization of this protein is of high interest the novelty of the structural information as presented here is rather low. The authors have already published a structure of the ATAD2 homolog from yeast, Abo1, which has a very similar organizational structure and fold. The Abo1 manuscript published by Cho et al elegantly connected the structure of Abo1 to its function via ATPase activity assays, histone loading assays, and HS-AFM to show that the ATPase activity was required to place histone H3-H4 onto naked DNA and that one of the 6 subunits was involved in the catalytic activity driving the molecular motor functions, and had a conformational change as ATP was utilized by this molecular motor. This established Abo1 as a chromatin remodeler involved in histone loading. Another homolog of ATAD2, Yta7 has since been shown to remove histones from the nucleosome. However, the function of human ATAD2 is currently unknown, and the current manuscript does little more than present the structure of ATAD2 as a hexameric complex. Beyond that the claims by the authors about the function of ATAD2 are not supported by the data presented. The MS data shown does indicate that ATAD2 interacts with the H3-H4 histones as one would expect, but the structural data presented to support the configuration of the bromodomain region and the histones is not of high enough quality/resolution to make any conclusions, and it is not clear that the regions identified represent the bromodomain or histone molecules since the sequence of the protein in these regions could not be determined. Thus, the descriptions of these interactions are highly speculative. Overall, there are several major flaws in the data presented that make this study unsuitable for publication.

> We thank the reviewer(s) for their helpful comments. We acknowledge that our structure of ATAD2 does not make advances in elucidating the function of ATAD2 and could be a potential weak point of our paper. However, as mentioned by Reviewer #2, our work proposes a novel auto-regulatory mechanism for ATAD2, and identifies the gate loops as the ISS motifs, providing important structural insights into ATAD2 regulation, and reveals structural nuances that differentiate human ATAD2 from other ATAD2 homologs. As suggested by reviewer #3, we have tried to improve figure clarity and provide more detailed methods and EM density so that these points might be clearer and more convincing. We have

also performed additional experiments to improve cryo-EM resolution and visualize conformational changes of ATAD2, which hopefully bolster our claims on the structural regulation of ATAD2.

1. The methods sections are incomplete at best, and in many cases completely absent. This raises questions about if the approach used, and if any artifacts were introduced that contribute to non-biological interactions. For example, there no methods presented for any of the data in the supplementary materials. This includes SEC chromatography (what column), how were molecular weights determined, for the negative stain images, what stain was used, and what protein concentrations, for the MS data there is NOTHING written to evaluate how the experiment was performed. The rigor and reproducibility of the experiments presented is not existent and not acceptable.

> We apologize that detailed methods were not provided for some of the material, and that all methods for both main and supplemental material were lumped into the main methods. We have now moved some of the information on gel filtration and XL-MS to the supplementary methods, and included information on the SEC molecular weight calibration and negative stain EM methods.

2. The lack of any ATPase activity in the ATAD2 protein construct they obtained is highly concerning. It appears the authors have not purified an active complex so much of the discussion that related to conformational changes during catalysis are not valid. As such, the authors have not determined if ATAD2 is a chaperone or chromatin remodeler and any reference to that need to be removed from the discussion. Furthermore, although this is an impressive structure at 3.1Å any regions of the structure that analyzes conformational changes, particularly hydrogen bond interactions, intersubunit contacts, and gate loop opening/closing need to be supported by density of that specific region. There is only one figure in the supp data that shows the quality of the local resolution and it isn't even stated what part of the structure/density is shown. The resolution of structures determined by CryoEM is highly variable from region to region. For example, the AAA1 ATPase domain has ~3Å resolution, but the bromodomain region has greater than 12Å resolution. It is not possible to describe bond interactions at such low resolution where secondary structure may not be discernable.

> Although the Walker B mutant used in our structural studies does not have wild type catalytic activity, we believe that the structure of the Walker B mutant likely reflects the structural principles of wild type ATAD2 for several reasons. First, Walker B mutants have been commonly used for studies of AAA+ ATPases such as Yme1 (Puchades et al, 2017), Msp1 (Wang et al, 2020), and Rix7 (Lo et al, 2019). In these studies, predictions made from structures of the Walker B mutant were validated in biochemical assays (Yme1 and Msp1), or in in vivo studies (Rix7, a case where the wild type protein was unattainable similar to our study), suggesting that lessons learned from Walker B mutants translate directly to wild type proteins. Moreover, in cases where both structures of the wild type and Walker B mutants were available (Msp1), the differences in conformation were minor and reflected the same mechanism. Second, although the Walker B mutant is usually considered a "catalytically dead" mutant, our previous observations of Abo1 by HS-AFM show that the Walker B mutant is not completely "dead", but actually undergoes a single symmetry breaking conformational change upon binding and hydrolysis of ATP. The only difference between wild type and Walker B mutant Abo1 is that wild type protein can undergo multiple cycles of symmetry breaking, while Walker B mutant can only undergo a single symmetry breaking event and gets "stuck" in the asymmetric state. Based on these observations, we also observed ATAD2 Walker B mutant by HS-AFM and found that symmetry breaking did occur, although less frequently and less prominently than Abo1. This suggests that the ATAD2 Walker B mutant retains aspects of conformational change that are expected to be observed in wild type ATAD2.

> With regards to the catalytic activity and function of ATAD2, we unfortunately do not gain any direct insights into these aspects from our current study, and do not make any claims as such, but hope to address these questions in a future study.

> To address the reviewer's questions about "real" resolution and the reliability of our interpretations in the ATAD2 Walker B map, we have created a new Supplementary figure (**new Supp Fig. 4**) that shows good map-model agreement at many scales and regions, and also includes density maps for the gate loops, substrate, and the N-terminal linker domain. We show that the regions where we propose detailed interactions and conformational changes all lie in the AAA rings, and that the "real resolution" of this region is reliably in the 3-4 Å range.

> Regarding the reviewer's point that there are local regions of low resolution in the ATAD2-H3/H4 complex structure, we agree that it is not possible to describe bond interactions, backbone conformation, etc, at this resolution, and did not make any direct claims with regards to those regions. We did make some tentative claims about the identity of these low resolution regions, (please also refer to rebuttal comments #6 &7 to Reviewer #2, and comments on Fig. 6 below), but have removed or toned down some of these claims based on the low resolution.

3. The figures are poorly labeled and confusing to follow. The descriptions and labeling need to be improved. For example:

- Figure 1B should have scale bars for the 2D classes. The model in Fig 1e, the linker arm regions (dotted lines) don't accurately represent the amino acid lengths. Also, the bromodomain in this figure should be greyed out, there is no evidence in structures presented in figure one for where the bromodomain is located with respect to the AAA domains.

> We have added scale bars for the 2D classes and 3D maps in Fig 1. Fig. 1e is intended to be a schematic that represents the connectivity and approximate 3-dimensional relationship of the different domains of ATAD2. We have represented the bromodomain as a dotted line as suggested, and added a caveat in the Fig. 1 legend that the figure is a schematic, where the dotted lines represent connectivity, but not necessarily amino acid lengths. We have also explained our rationale (parallels to the Abo1 structure, and extra density visible near the top surface in 2D class averages) for drawing the bromodomain near the top surface of the AAA ring on **pg. 8 (lines 178-181)**.

- Figure 2 D, E, f, g more labelling is needed. It is not clear which structures are ATAD2 vs Abo1. Abbreviations should be spelled out in the figure legend. Color coding for the first and second AAA domains would clarify D-E, or labeled. Things that are discussed in the manuscript text are not labeled in D-E, or other parts of this figure. Figure 2 would be improved by having top side and bottom views.

> In Fig. 2, we have added shaded columns that group figures from the same structure together and should help distinguish which structures are ATAD2 vs. Abo1. We have spelled out the abbreviations NBD: nucleotide binding domain and HBD: helical bundle domain in the figure legend, and added labels the knob-hole structures within the figure. We have also added top, side, and bottom views of the ATAD2 structure color-coded by chain in Fig. 1, which might help orient the readers in Fig. 2, and also adjusted the colors of the AAA1 HBD in Fig. 2d and e to help distinguish different domains.

Fig 3b no information is given for the sigma value of the density map. Also for the conformational changes to be supported the density needs to be shown for the gate loop region. Figures c, d and E are very difficult to follow or understand what is being presented. The color coding between subunits is not matching between the panels. P6 is pink in A and E, and Red and B, C, D.

> We have reported sigma values ($\sigma= 4.5$) for the density map in Fig. 3b, and also added density maps of all gate loops in Supplementary fig. 4. We have enlarged and rearranged the panels in Fig. 3 so that the

figure might be easier to follow. The “pink” color of P6 might be a result of color distortion during printing or an effect of transparency settings. We have adjusted the transparency setting so that the colors look more consistent in our view of the figures.

- Figure 4 need to show the density to support conformation and bond interactions of the LD region. Also a comparison of the density between other subunits would be helpful to compare the difference between the seams of P1-P6 vs P2-P3 for example.

- > We have added density maps of the LD and adjacent regions at the P1-P6 interface density in Supplementary fig. 4, and have juxtaposed with a similar view of the P2-P3 interface to compare the presence and absence of the LD.

- Figure 5, is difficult to follow and not labeled well, particularly C and D. The comparison between Abo1 and ATAD2 show much different views of the structure in A and B. The insert of the model in A, should include density around the model.

- > In Fig. 5c and d, we have added an inset indicating the enlarged regions shown in the figures, as well as adding more labels. The density map of the substrate in Fig. 5a is shown in Supplementary fig. 4c.

- Figure 6. The density supporting placement of the bromodomain and histone is low resolution or poor quality such that no sequence information can be derived and it is unclear that the density represents these proteins/regions. The MS data to support the interactions has no method information, and it is quite possible that the crosslinking performed to establish the ATAD2-histone complex created non-natural configurations, or are not the proteins expected. The bromodomain model does not look like it fits well into their density, particularly bromodomain 2. More detailed figures in the supp data may help.

- > We agree that placement of the bromodomain(s) might be premature (Please also refer to Reviewer #2 comment #6) considering the low resolution of the ATAD2-H3/H4 structure in the original manuscript. Thus, we have removed any discussion of domain conformation of the “handle class” in the revised manuscript, and have only shown the cryo-EM map of this class in Supplementary fig. 12.

- > However, during revision, we were able to obtain a higher-resolution structure of the ATAD2-H3/H4 structure (corresponding to “Class II - central density class”, which shows well-resolved density of the ATAD2 AAA+ domains, and the extra central density connecting to a substrate peptide in the central pore. This together with the XL-MS data increases confidence in the idea that a histone H3/H4 substrate is docked to the central pore via the H3 N-terminal tail. In addition, due to the higher resolution of the ATAD2-H3/H4 structure, we could also perform a more thorough comparative analysis of ATAD2 Walker B vs ATAD2 Walker B - H3/H4 complex, and re-confirm key conformational differences at the seam.

- > Regarding the MS data, we have moved and edited the methods that were in the main methods to supplementary methods, and provided additional supplementary data to show that products of crosslinking are likely similar to native products. Please also note that crosslinking and fractionation conditions for XL-MS and cryo-EM were equivalent, which increases our confidence that the crosslinks detected by MS are not non-specific crosslinks in non-natural configurations.

In summary, while the cryoEM structure of ATAD2 is important to study many of the claims made in the manuscript are unfounded based on the data presented. This is particularly true for the function of LD domain, the ATAD2-histone interactions, and the position of the bromodomain. The lack of detailed and missing methods is extremely concerning. As such, the study is not acceptable for publication.

- > We have added detailed density maps for high-resolution regions of the ATAD2 structure, improved the resolution of the ATAD2-H3/H4 structure, and removed some more tentative claims (such as the position of the bromodomain) about the low- resolution regions of the ATAD2-H3/H4 structure. We have also added detailed methods that should improve the reproducibility and credibility of the study.

Additional specific comments:

- In the line 334 authors have found ATAD2 interaction with histone H3/H4 dimer but from the material method section line 505 it looks like they used histone H3/H4 tetramer /

> It is well established that histone H3/H4 exists in a dimer/tetramer equilibrium, where H3/H4 exists mostly as a stable tetramer at high salt concentrations > 1 M, and as dimers at physiological salt concentrations (150 mM salt) (Donham et al, 2011).

Because we dilute H3/H4 from high salt to low salt when we mix with Abo1, we think that H3/H4 should initially exist predominantly as dimers. We cannot definitively determine whether H3/H4 exists as a dimer in the ATAD2 H3/H4 complex, but speculate that this is the most plausible model based on several lines of reasoning (please refer to comments below regarding binding stoichiometry and dimer/tetramer conformation).

- It will be helpful for the researchers in the field to mention in the discussion section about the difficulties/reasons of not obtaining a good resolution for ATAD2/H3-H4 complex structure compared to ATAD2 alone.

> This is an interesting point that also relates to the following point the reviewer has raised. Although we have improved the structure of the ATAD2-H3/H4 complex during revision, we think that the lower resolution of the ATAD2-H3/H4 complex compared to ATAD2 is probably an inherent structural difference related to the fact that ATAD2-H3/H4 has increased heterogeneity compared to ATAD2. In the ATAD2 alone structure, the hexameric ring is relatively planar with the seam fixed by the N-terminal LD, thus resulting in a highly homogeneous particles and a high-resolution structure. However, for the ATAD2-H3/H4 complex, we observe asymmetric hexamers, where the seam subunit is significantly disordered, and the N-terminal LD is partially undocked from the seam. This suggests that binding of H3/H4 to ATAD2 induces a more asymmetric and dynamic conformation. We have added related comments on **pg. 21 (lines 482-487)** of the discussion.

- Line 322-325 states in the class 1 (ATAD2 alone at 4.7 Å) authors have found the extra density for bromodomain. When they tried a cryo-EM study with ATAD2 + ATP alone they achieved a better resolution of 3.1 Å but no density for bromodomain was observed. State any specific reason in the discussion section of not obtaining a density for bromodomain at 3.1 Å structure.

> The reason we do not obtain density for the bromodomains in the 3.1 Å structure of ATAD2 alone is likely because of the flexibility of the bromodomains with respect to the AAA domains. We observe similar phenomena in the Abo1 structure (unpublished data), where the bromodomains are of much lower resolution than the AAA domains, although resolution in the case of Abo1 is sufficient to identify secondary structures of the bromodomains. We speculate that we were able to identify density for bromodomains in the 4.7 Å structure of the ATAD2-H3/H4 complex because H3/H4 binding might stabilize some bromodomains and decrease flexibility.

- Line 307-311 states that AAA1, bromo and C-terminal linker domain of ATAD2 interacts with histone H3/H4 and Line 334 states ATAD2 binds to one H3/H4 dimer. ATAD2 being hexamer then why it accommodates only one H3/H4 dimer why not six? I would suggest including this point in the discussion section.

> Based on our cryo-EM data alone we cannot definitively determine the binding stoichiometry of ATAD2 and histone H3/H4. However, ATAD2 binds the H3 tail by the AAA pore similar to Abo1, an extra density connecting to the AAA pore is observed in the ATAD2-H3/H4 complex structure, suggesting that an H3/H4 dimer binds to the AAA ring by the AAA pore. Since the ATAD2 hexamer has a single pore that can only accommodate a single H3 tail, we favor the idea that the ATAD2 binds an H3/H4 dimer. Our

work on Abo1 binding to H3/H4 (manuscript in preparation) also suggests that an H3/H4 dimer binds to the Abo1 hexamer. In contrast, we disfavor the idea that an ATAD2 hexamer binds 6 histone H3/H4 molecules as we do not observe significant shifts in native gels upon H3/H4 binding. We have added related comments on **pg. 22 (lines 519-522)**.

- I would also recommend doing some additional biophysical experiment such AUC or SAXS to conform the ratio and binding of ATAD2/H3-H4.

- > Unfortunately, we think that AUC and SAXS would not provide the resolution to confirm the stoichiometry of H3/H4 binding to ATAD2. ATAD2 has a molecular weight of ~700kDa, and H3/H4 20kDa. With AUC experiments, it would be difficult to distinguish among ATAD2 (700kDa), ATAD2-H3H4 (720kDa), and ATAD2-(H3H4)₂ (740kDa) complexes. Also, the resolution of SAXS would likely be ~10Å at best, and based on the high disorder we see in the bromodomains and substrate binding regions of the ATAD2-H3/H4 complex, we expect that it would be difficult to obtain reasonable reconstructions of H3/H4 and the H3/H4 binding regions. We are exploring other avenues to better characterize the ATAD2-H3/H4 interaction, but this has not been straightforward, and we plan to reserve these experiments for a future study.

- Line 509-512 states that ATAD2 and H3/H4 was incubated at 150 mM NaCl but histone H3/H4 tetramer was purified at 2M NaCl. Does lower of the salt affect the stability of histone H3/H4 tetramer if not then how the salt conc. was decreased for the binding reaction can be written in detail for replicating the result by other researchers in the field.

- > As mentioned in the comment above, high salt (2 M NaCl) stabilizes the histone H3/H4 tetramer, and is the standard condition at which histones are refolded and stored for biochemical experiments (Luger et al, 1999). Histone H3/H4 exists in a dimer/tetramer equilibrium, where higher salt favors the formation of tetramers. To obtain ATAD2-H3/H4 complexes, a concentrated stock of refolded H3/H4 stored in 2 M NaCl was first diluted into a 150mM NaCl buffer, and then subsequently incubated with ATAD2. We have added this phrase to the methods (**pg. 27 lines 629-633**) to clarify these details.

- Line 515-517 states that author have used sucrose gradient for the purification of ATAD2/H3-H4 crosslinked sample. I would suggest adding the SDS-PAGE gel image in the of the fractions obtained from the sucrose gradient as supplementary figures for better clarity.

- > We thank you for the suggestion, and have now added this information to **new Supplementary fig. 8**.

Reviewers' comments:

Reviewer #2 (Remarks to the Author):

The authors have made significant changes and improved the overall quality of the manuscript. All major and minor questions and comments have been addressed. Thank you for considering my suggestions and congratulations on the nice work.

Reviewer #3 (Remarks to the Author):

Review summary:

The manuscript by Cho et al. presents the structure of ATAD2, with a focus on its interaction with histone ligands using CryoEM. In this revised version, the authors have made significant improvements, particularly in the writing, methods, and figure presentation. However, I have some concerns regarding the overall impact and significance of the findings.

The authors have succeeded in obtaining a high-quality structure of ATAD2, which is commendable. The improved figures and detailed legends contribute to a better understanding of the ATAD2 structure and its features. Nevertheless, the observation that ATAD2 shares high structural conservation with Abo1 and Yta7 is somewhat expected and may limit the novelty of the structure alone.

A major concern lies in the lack of AAA ATPase activity in ATAD2, which hinders functional studies. This limitation significantly weakens the impact of the structure as it prevents a comprehensive understanding of ATAD2's role in cellular processes.

The most critical issue is the questionable resolution of ATAD2 in complex with histone ligands. The absence of density for histones raises doubts about the validity of the claimed changes in the structure due to a ligand interaction. It is essential for the authors to reevaluate the quality of the density maps, and employ gold-standard practices in data processing to confirm the resolution and validity of their findings. Without such confirmation, the claims of ligand interaction may be premature and could be attributed to overfitting or other artifacts.

Considering the current state of the manuscript, I share the reviewer's opinion that a single CryoEM structure without functional data may not meet the standards required for publication in high-impact journals like Nature Communications. To strengthen the manuscript's scientific significance, I strongly suggest revisiting the functional aspects and resolving the concerns regarding the ligand interaction through additional experiments and rigorous data analysis. In summary, while the improvements in writing, methods, and figures are commendable, the manuscript's overall impact and significance are weakened by the lack of functional studies and concerns about the resolution of ATAD2 in complex with histone ligands. I recommend major revisions that address these issues before considering the manuscript for publication in a reputable journal like Nature Communications.

Major concerns:

1. In Supplementary Figure 1b, there is no molecular weight (MW) marker included in the gel, and the predicted/expected MW values for each construct are missing. This makes it challenging to determine the elution position of the bands observed in the SEC column in Supplementary Figure 1a.
2. Supplementary Figure 2 lacks important information about the data processing workflow, such as whether homogeneous or heterogeneous refinement was performed, and the details of masking applied. Additionally, presenting the FSC curves with a graph that includes the resolution in angstroms

directly (rather than $1/\text{\AA}$) would help readers evaluate the final resolution easily. Best practice would be to show solid lines at the FSC cutoff values of 0.143 and 0.5 to gauge the true resolution of the reconstructions from all particles.

3. The claim of additional density for the bromodomain on the top surface of the ATAD2 hexamer in the class averages from Figure 1b and Supplementary Figure 2 lacks supporting evidence. It is recommended to remove the last sentence of the paragraph preceding the 'hexameric spiral assembly of ATAD2' and edit Figure 1f to remove the bromodomain.

4. Supplementary Figure 10's FSC curve indicates an overestimation of the resolution of the map, possibly due to overfitting during post-processing. A detailed description of how the FSC curve was calculated is needed, and following gold standard practices is recommended (Scheres and Chen 2012). The authors should also include cutoff lines at 0.148 and 0.5 for the final highest resolution and combined resolution of all particles, respectively. The 'crunchy' appearance of the final density maps raises concerns about overfitting into noise, and the authors need to ensure the validity of the structure by resolving these issues.

5. The current data does not support the statement that histone H3/H4 binding causes the undocking of the N-terminal LD, potentially unleashing ATAD2 activity and triggering nucleotide-dependent dynamics. The resolution in the structure is insufficient to support this claim, and further experiments are needed.

6. It is unclear from the current study if histones H3/H4 are the biological substrates of ATAD2. The approach of incubating them together and then crosslinking them does not prove that they naturally interact. The structure of ATAD2 in complex with histones shows no ordered location for the histones (there is no density for them).

7. The XL-MS studies show the N-term tail of histone H3 and H4 interacting with the bromodomain, but only acetylated H4 has been shown to bind the bromodomain, so it is unclear why the bromodomain would bind to H4 in this experiment where no modifications are present on the histones. Instead, the ATAD2-histone XL-MS data likely represent non-natural interactions induced by the crosslinking conditions. Abo1 bound tightly to the histones (nM affinity), and a similar affinity measurement should be taken for the ATAD2-histone interaction. Also, the XL-MS data does not include specific residue interactions observed so it is not possible to evaluate if the interactions observed are biologically relevant.

8. The lower resolution observed in the ATAD2-histone complex is likely due to the significantly lower number of particles used in structure determination rather than the histone interaction causing a conformational change. Claims regarding the structural change and lower resolution need to be reevaluated.

9. The density maps in Supplementary Figures 10, 11, 12, and 13 are of poor quality and do not accurately represent protein secondary structure. These maps may be the result of noise from overfitting and overestimation of resolution, casting doubts on the proposed model.

10. The AFM data do not align with the proposed model regarding histone-induced allosteric effects on ATAD2. The molecular mechanism underlying the observed dynamic conformational changes requires further investigation.

In conclusion, the manuscript requires substantial revisions to address the major concerns raised. Additional experiments, data processing improvements, and proper validation of resolution are essential before considering the publication of the ATAD2-histone complex structures.

Minor concerns:

1. In Supplementary Figure 1, the protein name descriptions lack clarity. Please include the amino acid residue ranges of each construct to make it more explicit.

2. There are several typos present in both the main document and the supplemental data text, such as "FSC" not being in all caps, missing "histone" before "H3/H4," using "AAA" instead of "AAA+," and

lacking spaces between numbers and units or "Table" and "1." These errors should be corrected for accuracy and consistency.

3. When comparing with published structures, it would be helpful to include the PDB ID numbers for those structures in the text and figure legends (e.g., Msp1, katanin, unfoldases, p97, etc.).

Additionally, references should be provided for these solved structures to support the comparisons made.

4. The references for the Abo1 and Yta7 studies are missing in the discussion section. Ensure that appropriate citations are included to give credit to the original authors and provide readers with access to the relevant literature.

5. Throughout the manuscript, the nomenclature for N-terminally truncated ATAD2 (aa 403-1390) is inconsistent, especially in the supplemental methods. Ensure that the correct nomenclature is used consistently throughout the text.

6. Please include references that support the interaction of the ATAD2 bromodomain with H4K5 acetylated histones. Properly citing relevant studies will strengthen the validity of this claim.

Addressing these minor concerns will improve the clarity, accuracy, and overall quality of the manuscript and supplemental data.

Response to Reviewers' comments:

Reviewer #2 (Remarks to the Author):

The authors have made significant changes and improved the overall quality of the manuscript. All major and minor questions and comments have been addressed. Thank you for considering my suggestions and congratulations on the nice work.

> We are glad to hear that the reviewer finds the revised manuscript improved. We sincerely thank the reviewer again for their critique and many constructive suggestions.

Reviewer #3 (Remarks to the Author):

Review summary:

The manuscript by Cho et al. presents the structure of ATAD2, with a focus on its interaction with histone ligands using CryoEM. In this revised version, the authors have made significant improvements, particularly in the writing, methods, and figure presentation. However, I have some concerns regarding the overall impact and significance of the findings.

The authors have succeeded in obtaining a high-quality structure of ATAD2, which is commendable. The improved figures and detailed legends contribute to a better understanding of the ATAD2 structure and its features. Nevertheless, the observation that ATAD2 shares high structural conservation with Abo1 and Yta7 is somewhat expected and may limit the novelty of the structure alone.

A major concern lies in the lack of AAA ATPase activity in ATAD2, which hinders functional studies. This limitation significantly weakens the impact of the structure as it prevents a comprehensive understanding of ATAD2's role in cellular processes.

The most critical issue is the questionable resolution of ATAD2 in complex with histone ligands. The absence of density for histones raises doubts about the validity of the claimed changes in the structure due to a ligand interaction. It is essential for the authors to reevaluate the quality of the density maps, and employ gold-standard practices in data processing to confirm the resolution and validity of their findings. Without such confirmation, the claims of ligand interaction may be premature and could be attributed to overfitting or other artifacts.

Considering the current state of the manuscript, I share the reviewer's opinion that a single CryoEM structure without functional data may not meet the standards required for publication in high-impact journals like Nature Communications. To strengthen the manuscript's scientific significance, I strongly suggest revisiting the functional aspects and resolving the concerns regarding the ligand interaction through additional experiments and rigorous data analysis. In summary, while the improvements in writing, methods, and figures are commendable, the manuscript's overall impact and significance are weakened by the lack of functional studies and concerns about the resolution of ATAD2 in complex with histone ligands. I recommend major revisions that address these issues before considering the manuscript for publication in a reputable journal like Nature Communications.

> We thank Reviewer #3 for their critique of our work. We have tried to clarify some of the issues raised below which we hope strengthens our claims. Regarding the ATPase activity, we have been trying for several years to obtain ATAD2 showing ATPase activity with many different constructs. However, we are unable to obtain ATAD2 protein showing ATPase activity so far. While we acknowledge that the lack of ATPase activity hinders further investigation of the function of ATAD2, this manuscript presents the

atomic resolution cryo-EM structure of human ATAD2, which is clinically relevant and provides structural insights into the mechanism of ATAD2 works in the presence of histones. In addition, this manuscript is being considered for publication in Communications Biology, NOT Nature Communications.

Major concerns:

1. In Supplementary Figure 1b, there is no molecular weight (MW) marker included in the gel, and the predicted/expected MW values for each construct are missing. This makes it challenging to determine the elution position of the bands observed in the SEC column in Supplementary Figure 1a.

> The gel in Supplementary Figure 1b is a native PAGE. Although native gels provide a general idea about the oligomeric state(s) and homogeneity of proteins, they do not provide a good estimate of absolute molecular weight because mobility can also be affected by various factors such as the shape or surface charge of the molecule. Thus, it is uncommon and non-informative to run molecular weight markers on native gels. Instead, the gel filtration markers run on the gel filtration column give a better approximation of the MW values of each construct, and we have indicated the expected molecular weight values of N-terminally truncated ATAD2 (696 kDa as a hexamer and 116 kDa as a monomer) in the Supplementary figure 1 legend.

2. Supplementary Figure 2 lacks important information about the data processing workflow, such as whether homogeneous or heterogeneous refinement was performed, and the details of masking applied. Additionally, presenting the FSC curves with a graph that includes the resolution in angstroms directly (rather than $1/\text{\AA}$) would help readers evaluate the final resolution easily. Best practice would be to show solid lines at the FSC cutoff values of 0.143 and 0.5 to gauge the true resolution of the reconstructions from all particles.

> The data in Supplementary Figure 2 was processed with Relion 3.1, where homogenous refinement is the only option. We also attempted to improve resolution by heterogeneous refinement in cryoSPARC but the resolution was worse than with 3D classification and homogeneous refinement in Relion. We have added the term “homogenous” cryo-EM refinement to the figure legend to clarify this issue. We have also added additional details of refinement and post-processing in the Methods, and revised the FSC curve to reflect the reviewer’s comments.

3. The claim of additional density for the bromodomain on the top surface of the ATAD2 hexamer in the class averages from Figure 1b and Supplementary Figure 2 lacks supporting evidence. It is recommended to remove the last sentence of the paragraph preceding the 'hexameric spiral assembly of ATAD2' and edit Figure 1f to remove the bromodomain.

> The position of the bromodomain in Figure 1f is a cartoon schematic and a tentative proposal of the position of the bromodomain. We believe that the proposal is plausible based on parallels to the Abo1/Yta7 structure and the position of the additional density that we observe in 2D class averages as well as the refined 3D model. In addition, please refer to note from the editor.

4. Supplementary Figure 10's FSC curve indicates an overestimation of the resolution of the map, possibly due to overfitting during post-processing. A detailed description of how the FSC curve was

calculated is needed, and following gold standard practices is recommended (Scheres and Chen 2012). The authors should also include cutoff lines at 0.148 and 0.5 for the final highest resolution and combined resolution of all particles, respectively. The 'crunchy' appearance of the final density maps raises concerns about overfitting into noise, and the authors need to ensure the validity of the structure by resolving these issues.

> The gold-standard FSC curve in Supplementary Figure 10 was obtained from Relion 4.0 after iterative auto-refinement and CTF refinement, followed by a final post-processing step, where a low pass-filtered and extended mask with a soft edge was used. We have added the cutoff lines to indicate FSC values of 0.143 and 0.5. As the dataset in Supplementary fig. 10 clearly shows heterogeneous classes in the dataset, combining all particles and reporting the highest resolution would not necessarily be informative as averaging heterogeneous classes would obviously drag down the resolution and worsen the quality of the maps.

> The “crunchy” appearance of the final density maps is likely related to the lower resolution of the non-AAA+ domains which we surmise results from the flexibility of the non-AAA+ domains with respect to the AAA+ domains. However, the map of the AAA+ domains (as shown in Supplementary fig. 12 and 13) is clearly near ~4 angstrom resolution, and is sufficient to support our claims on AAA+ structural change. (Please also see related response to comment #9).

5. The current data does not support the statement that histone H3/H4 binding causes the undocking of the N-terminal LD, potentially unleashing ATAD2 activity and triggering nucleotide-dependent dynamics. The resolution in the structure is insufficient to support this claim, and further experiments are needed.

> Our model of N-terminal LD undocking and ATAD2 activation is based on the comparison of cryo-EM structures of ATAD2 with or without histones. The resolution of the cryo-EM structures in the AAA+ ring is 3-4 angstroms, giving us confidence in the structural changes we observe in the N-terminal LD and AAA+ domains which are all part of the AAA+ ring. In addition, please refer to note from the editor.

6. It is unclear from the current study if histones H3/H4 are the biological substrates of ATAD2. The approach of incubating them together and then crosslinking them does not prove that they naturally interact. The structure of ATAD2 in complex with histones shows no ordered location for the histones (there is no density for them).

> There are a significant number of studies of ATAD2 and ATAD2 homologs that establish that histone H3/H4 is a biological substrate of ATAD2 including but not limited to the following papers cited in the manuscript: Caron et al (2010), Morozumi et al (2016), Chacin et al (2021), and Gal et al (2016). In addition, the fact that we see a significant number of crosslinks that are clustered in specific regions of both proteins adds confidence that our crosslinked ATAD2-H3/H4K5Q complexes are not non-specifically crosslinked complexes. In addition, please refer to the note from the editor.

7. The XL-MS studies show the N-term tail of histone H3 and H4 interacting with the bromodomain, but only acetylated H4 has been shown to bind the bromodomain, so it is unclear why the bromodomain would bind to H4 in this experiment where no modifications are present on the histones. Instead, the

ATAD2-histone XL-MS data likely represent non-natural interactions induced by the crosslinking conditions. Abo1 bound tightly to the histones (nM affinity), and a similar affinity measurement should be taken for the ATAD2-histone interaction. Also, the XL-MS data does not include specific residue interactions observed so it is not possible to evaluate if the interactions observed are biologically relevant.

> There are a significant number of studies of ATAD2 and ATAD2 homologs that establish that histone H3/H4 is a biological substrate of ATAD2 including but not limited to the following papers cited in the manuscript: Caron et al (2010), Morozumi et al (2016), Chacin et al (2021) Gal et al (2016). In addition, the fact that we see a significant number of crosslinks that are clustered in specific regions of both proteins adds confidence that our crosslinked ATAD2-H3/H4K5Q complexes are not non-specifically crosslinked complexes. We have added a table of the detected crosslinks in the XL-MS experiments as a supplementary data file for reference.

Regarding how ATAD2 can bind to unmodified histones, we must note that the six ATAD2 bromodomains are likely forced into a hexamer due to the hexameric ATPase base. When six bromodomains accommodate histone H3/H4, it is plausible that one of the six bromodomains might preferentially recognize the acetylated H4 tail, but that the other bromodomains might form multiple weak interactions with histones conferring an overall tight interaction between histones and ATAD2.

8. The lower resolution observed in the ATAD2-histone complex is likely due to the significantly lower number of particles used in structure determination rather than the histone interaction causing a conformational change. Claims regarding the structural change and lower resolution need to be reevaluated.

> Although the final number of particles in the ATAD2-histone complex structure is lower than that of the structure of ATAD2 alone, we disagree that “particle number” is the major reason for the differences between the ATAD2 alone and ATAD2-histone complex structures. We instead observed that there were significant differences between the ATAD2 and ATAD2-histone datasets during cryo-EM data processing. For the ATAD2 alone dataset, 3D classification did not yield any other significant classes besides the highly symmetric hexamer. Classification only yielded 1 good class with all other classes containing just “bad particles” with significantly lower resolution. However, for the ATAD2-histone dataset, 3D classification yielded several distinct classes with comparable resolution. This likely reflects the higher mobility and symmetry breaking frequency of ATAD2 in the histone complex state, and is likely an inherent reason for the lower resolution of the ATAD2-H3/H4 complex. In addition, please refer to the note from the editor.

9. The density maps in Supplementary Figures 10, 11, 12, and 13 are of poor quality and do not accurately represent protein secondary structure. These maps may be the result of noise from overfitting and overestimation of resolution, casting doubts on the proposed model.

> We acknowledge that the density outside of the AAA+ domains are of lower resolution, and do not claim to distinguish secondary structures from these maps. However, we do obtain sufficiently high resolution for the AAA+ rings, enabling us to resolve secondary structures and the protein backbone. Our proposed model is based on structural changes within the AAA+ ring, and the limited resolution outside of the AAA+ rings do not affect our conclusions. The poorer density outside of the AAA+ rings is

indicative of a protein structure with heterogeneous conformations, and likely reflects the flexibility of the non-AAA+ domains with respect to the AAA+ domains.

10. The AFM data do not align with the proposed model regarding histone-induced allosteric effects on ATAD2. The molecular mechanism underlying the observed dynamic conformational changes requires further investigation.

> Our AFM data shows increased ATAD2 AAA+ symmetry breaking in the presence of histones, and supports our proposal of histone-induced allosteric effects. Although additional studies might be required to further bolster our claims, we believe that the increased dynamics of the AAA+ ring aligns with our cryo-EM observations showing N-terminal LD undocking and symmetry breaking. In addition, please refer to the note from the editor.

In conclusion, the manuscript requires substantial revisions to address the major concerns raised. Additional experiments, data processing improvements, and proper validation of resolution are essential before considering the publication of the ATAD2-histone complex structures.

Minor concerns:

1. In Supplementary Figure 1, the protein name descriptions lack clarity. Please include the amino acid residue ranges of each construct to make it more explicit.

> All proteins used for the figures in this study are N-terminally truncated, consisting of aa 403-1390. The data represented show comparisons of N-terminally truncated proteins with or without the Walker B (E532Q) or N-terminal LD (D415A/R540A) mutation. We have revised the Supplementary fig. 1 legend and added a comment on pg. 6 “We therefore refer to N-terminally truncated ATAD2 as “ATAD2” or “wild type ATAD2” hereafter, and any point mutations introduced are in the N-terminally truncated ATAD2 (aa 403-1390) background”, to clarify the amino acid composition of the protein constructs used in this study.

2. There are several typos present in both the main document and the supplemental data text, such as "FSC" not being in all caps, missing "histone" before "H3/H4," using "AAA" instead of "AAA+," and lacking spaces between numbers and units or "Table" and "1." These errors should be corrected for accuracy and consistency.

> We have corrected for the aforementioned errors including using “AAA+” instead of “AAA” for consistency, correcting spacing between number and units and “Table” and “1” on pg. 8. and several Supplemental figures, and correcting “Fsc” to “FSC” in Supplemental figures 2 and 10.

> In the chromatin field, histones are commonly referred to just by their names, such that histone H3/H4 is referred to as H3/H4. We have chosen to use “H3/H4” instead of the full name “histone H3/H4” for brevity in cases where it is clear that H3/H4 refers to the histone H3/H4. However, we also used “histone H3/H4” when necessary, such as in the beginning of sections.

3. When comparing with published structures, it would be helpful to include the PDB ID numbers for

those structures in the text and figure legends (e.g., Msp1, katanin, unfoldases, p97, etc.). Additionally, references should be provided for these solved structures to support the comparisons made.

> We originally included the PDB ID numbers for Msp1 and katanin in the Figure 4b legend, and the references for these structures on pg. 11 (references #36 and #44-46). We also included the PDB ID numbers in the Fig. 5d legend for spastin and NSF. We have now added additional references for spastin and NSF on pg. 13 (references #46 and 47).

4. The references for the Abo1 and Yta7 studies are missing in the discussion section. Ensure that appropriate citations are included to give credit to the original authors and provide readers with access to the relevant literature.

> We have added the references for Abo1 and Yta7 in the discussion (on pg. 22)

5. Throughout the manuscript, the nomenclature for N-terminally truncated ATAD2 (aa 403-1390) is inconsistent, especially in the supplemental methods. Ensure that the correct nomenclature is used consistently throughout the text.

> We have clarified by mentioning in the main text, that after pg. 6 all experimental data was performed with N-terminally truncated ATAD2, and that “wild-type” hereafter refers to N-terminally truncated ATAD2.

6. Please include references that support the interaction of the ATAD2 bromodomain with H4K5 acetylated histones. Properly citing relevant studies will strengthen the validity of this claim.

> We provided citations for the acetylated histone binding preference of ATAD2 (on pg. 4), and also added references on pg. 14 to justify our rationale for producing ATAD2-H4K5Q complexes. We have also added the references for histone acetylation preference in the discussion section (on pg. 22).

Addressing these minor concerns will improve the clarity, accuracy, and overall quality of the manuscript and supplemental data.